# Deterministic Variational Inference for Robust Bayesian Neural Networks

**Anqi Wu**[1]*, **Sebastian Nowozin**[2]†, **Edward Meeds**[4],
**Richard E. Turner**[3,4], **José Miguel Hernández-Lobato**[3,4] &
**Alexander L. Gaunt**[4]
[1] Princeton Neuroscience Institute, Princeton University
[2] Google AI Berlin
[3] Department of Engineering, University of Cambridge
[4] Microsoft Research, Cambridge
anqiw@princeton.edu, nowozin@google.com,
{ret26,jmh233}@cam.ac.uk,{ted.meeds, algaunt}@microsoft.com

## Abstract

Bayesian neural networks (BNNs) hold great promise as a flexible and principled solution to deal with uncertainty when learning from finite data. Among approaches to realize probabilistic inference in deep neural networks, *variational Bayes* (VB) is theoretically grounded, generally applicable, and computationally efficient. With wide recognition of potential advantages, why is it that variational Bayes has seen very limited practical use for BNNs in real applications? We argue that variational inference in neural networks is fragile: successful implementations require careful initialization and tuning of prior variances, as well as controlling the variance of Monte Carlo gradient estimates. We provide two innovations that aim to turn VB into a robust inference tool for Bayesian neural networks: *first*, we introduce a novel deterministic method to approximate moments in neural networks, eliminating gradient variance; *second*, we introduce a hierarchical prior for parameters and a novel Empirical Bayes procedure for automatically selecting prior variances. Combining these two innovations, the resulting method is highly efficient and robust. On the application of heteroscedastic regression we demonstrate good predictive performance over alternative approaches.

## 1 Introduction

Bayesian approaches to neural network training marry the representational flexibility of deep neural networks with principled parameter estimation in probabilistic models. Compared to "standard" parameter estimation by maximum likelihood, the Bayesian framework promises to bring key advantages such as better uncertainty estimates on predictions and automatic model regularization (MacKay, 1992; Graves, 2011). These features are often crucial for informing downstream decision tasks and reducing overfitting, particularly on small datasets. However, despite potential advantages, such Bayesian neural networks (BNNs) are often overlooked due to two limitations: *First*, posterior inference in deep neural networks is analytically intractable and approximate inference with Monte Carlo (MC) techniques can suffer from crippling variance given only a reasonable computation budget (Kingma et al., 2015; Molchanov et al., 2017; Miller et al., 2017; Zhu et al., 2018). *Second*, performance of the Bayesian approach is sensitive to the choice of prior (Neal, 1993), and although we may have *a priori* knowledge concerning the *function* represented by a neural network, it is generally difficult to translate this into a meaningful prior on neural network weights. Sensitivity to priors and initialization makes BNNs non-robust and thus often irrelevant in practice.

In this paper, we describe a novel approach for inference in feed-forward BNNs that is simple to implement and aims to solve these two limitations. We adopt the paradigm of variational Bayes (VB) for BNNs (Hinton & van Camp, 1993; MacKay, 1995c) which is normally deployed using Monte

---

*Work done during an internship at Microsoft Research, Cambridge.
†Work done while at Microsoft Research, Cambridge.

Carlo variational inference (MCVI) (Graves, 2011; Blundell et al., 2015). Within this paradigm we address the two shortcomings of current practice outlined above: *First*, we address the issue of high variance in MCVI, by reducing this variance to zero through novel deterministic approximations to variational inference in neural networks. *Second*, we derive a general and robust *Empirical Bayes* (EB) approach to prior choice using hierarchical priors. By exploiting conjugacy we derive data-adaptive closed-form variance priors for neural network weights, which we experimentally demonstrate to be remarkably effective.

Combining these two novel ingredients gives us a performant and robust BNN inference scheme that we refer to as "deterministic variational inference" (DVI). We demonstrate robustness and improved predictive performance in the context of non-linear regression models, deriving novel closed-form results for expected log-likelihoods in homoscedastic and heteroscedastic regression (similar derivations for classification can be found in the appendix).

Experiments on standard regression datasets from the UCI repository, (Dheeru & Karra Taniskidou, 2017), show that for identical models DVI converges to local optima with better predictive log-likelihoods than existing methods based on MCVI. In direct comparisons, we show that our Empirical Bayes formulation automatically provides better or comparable test performance than manual tuning of the prior and that heteroscedastic models consistently outperform the homoscedastic models.

Concretely, our contributions are:

- Development of a deterministic procedure for propagating uncertain activations through neural networks with uncertain weights and ReLU or Heaviside activation functions.
- Development of an EB method for principled tuning of weight priors during BNN training.
- Experimental results showing the accuracy and efficiency of our method and applicability to heteroscedastic and homoscedastic regression on real datasets.

## 2 Variational Inference in Bayesian Neural Networks

We start by describing the inference task that our method must solve to successfully train a BNN. Given a model $\mathcal{M}$ parameterized by weights $\boldsymbol{w}$ and a dataset $\mathcal{D} = (\boldsymbol{x}, \boldsymbol{y})$, the inference task is to discover the posterior distribution $p(\boldsymbol{w}|\boldsymbol{x}, \boldsymbol{y})$. A variational approach acknowledges that this posterior generally does not have an analytic form, and introduces a variational distribution $q(\boldsymbol{w}; \boldsymbol{\theta})$ parameterized by $\boldsymbol{\theta}$ to approximate $p(\boldsymbol{w}|\boldsymbol{x}, \boldsymbol{y})$. The approximation is considered optimal within the variational family for $\boldsymbol{\theta}^*$ that minimizes the Kullback-Leibler (KL) divergence between $q$ and the true posterior.

$$\boldsymbol{\theta}^* = \underset{\boldsymbol{\theta}}{\operatorname{argmin}} \, D_{\mathrm{KL}} \left[ q(\boldsymbol{w}; \boldsymbol{\theta}) || p(\boldsymbol{w}|\boldsymbol{x}, \boldsymbol{y}) \right].$$

Introducing a prior $p(\boldsymbol{w})$ and applying Bayes rule allows us to rewrite this as optimization of the quantity known as the evidence lower bound (ELBO):

$$\boldsymbol{\theta}^* = \underset{\boldsymbol{\theta}}{\operatorname{argmax}} \left\{ \mathbb{E}_{\boldsymbol{w} \sim q} \left[ \log p(\boldsymbol{y}|\boldsymbol{w}, \boldsymbol{x}) \right] - D_{\mathrm{KL}} \left[ q(\boldsymbol{w}; \boldsymbol{\theta}) || p(\boldsymbol{w}) \right] \right\}. \tag{1}$$

Analytic results exist for the KL term in the ELBO for careful choice of prior and variational distributions (e.g. Gaussian families). However, when $\mathcal{M}$ is a non-linear neural network, the first term in equation 1 (referred to as the reconstruction term) cannot be computed exactly: this is where MC approximations with finite sample size $S$ are typically employed:

$$\mathbb{E}_{\boldsymbol{w} \sim q} \left[ \log p(\boldsymbol{y}|\boldsymbol{w}, \boldsymbol{x}) \right] \approx \frac{1}{S} \sum_{s=1}^{S} \log p(\boldsymbol{y}|\boldsymbol{w}^{(s)}, \boldsymbol{x}), \quad \boldsymbol{w}^{(s)} \sim q(\boldsymbol{w}; \boldsymbol{\theta}). \tag{2}$$

Our goal in the next section is to develop an explicit and accurate approximation for this expectation, which provides a deterministic, closed-form expectation calculation, stabilizing BNN training by removing all stochasticity due to Monte Carlo sampling.

## 3 Deterministic Variational Approximation

Figure 1 shows the architecture of the computation of $\mathbb{E}_{\boldsymbol{w} \sim q} \left[ \log p(\mathcal{D}|\boldsymbol{w}) \right]$ for a feed-forward neural network. The computation can be divided into two parts: first, propagation of activations though

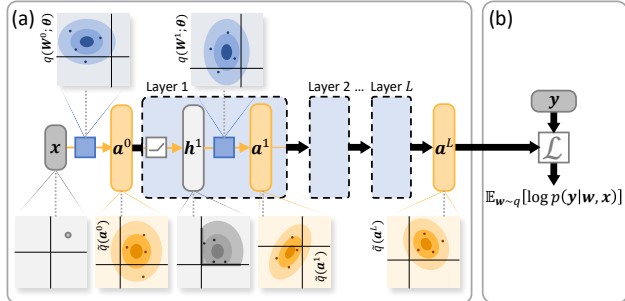

Figure 1: Architecture of a Bayesian neural network. Computation is divided into (a) propagation of activations ($\boldsymbol{a}$) from an input $x$ and (b) computation of a log-likelihood function $\mathcal{L}$ for outputs $y$. Weights are represented as high dimensional variational distributions (blue) that induce distributions over activations (yellow). MCVI computes using samples (dots); our method propagates a full distribution.

parameterized layers and second, evaluation of an unparameterized log-likelihood function ($\mathcal{L}$). In this section, we describe how each of these stages is handled in our deterministic framework.

## 3.1 MOMENT PROPAGATION

We begin by considering activation propagation (figure 1(a)), with the aim of deriving the form of an approximation $\tilde{q}(\boldsymbol{a}^L)$ to the final layer activation distribution $q(\boldsymbol{a}^L)$ that will be passed to the likelihood computation. We compute $\boldsymbol{a}^L$ by sequentially computing the distributions for the activations in the preceding layers. Concretely, we define the action of the $l^{\text{th}}$ layer that maps $\boldsymbol{a}^{(l-1)}$ to $\boldsymbol{a}^l$ as follows:

$$\boldsymbol{h}^l = f(\boldsymbol{a}^{(l-1)}),$$
$$\boldsymbol{a}^l = \boldsymbol{h}^l \boldsymbol{W}^l + \boldsymbol{b}^l,$$

where $f$ is a non-linearity and $\{\boldsymbol{W}^l, \boldsymbol{b}^l\} \subset \boldsymbol{w}$ are random variables representing the weights and biases of the $l^{\text{th}}$ layer that are assumed independent from weights in other layers. For notational clarity, in the following we will suppress the explicit layer index $l$, and use primed symbols to denote variables from the $(l-1)^{\text{th}}$ layer, e.g. $\boldsymbol{a}' = \boldsymbol{a}^{(l-1)}$. Note that we have made the non-conventional choice to draw the boundaries of the layers such that the linear transform is applied after the non-linearity. This is to emphasize that $\boldsymbol{a}^l$ is constructed by linear combination of many distinct elements of $\boldsymbol{h}'$, and in the limit of vanishing correlation between terms in this combination, we can appeal to the central limit theorem (CLT). Under the CLT, for a large enough hidden dimension and for variational distributions with finite first and second moments, elements $a_i$ will be normally distributed regardless of the potentially complicated distribution for $h_j$ induced by $f$[1]. We empirically observe that this claim is approximately valid even when (weak) correlations appear between the elements of $\boldsymbol{h}$ during training (see section 3.1.1).

Having argued that $\boldsymbol{a}$ adopts a Gaussian form, it remains to compute the first and second moments. In general, these cannot be computed exactly, so we develop an approximate expression. An overview of this derivation is presented here with more details in appendix A. First, we model $\boldsymbol{W}$, $\boldsymbol{b}$ and $\boldsymbol{h}$ as independent random variables, allowing us to write:

$$\langle a_i \rangle = \langle h_j \rangle \langle W_{ji} \rangle + \langle b_i \rangle,$$
$$\text{Cov}(a_i, a_k) = \langle h_j h_l \rangle \text{Cov}(W_{ji}, W_{lk}) + \langle W_{ji} \rangle \text{Cov}(h_j, h_l) \langle W_{lk} \rangle + \text{Cov}(b_i, b_k), \quad (3)$$

where we have employed the Einstein summation convention and used angle brackets to indicate expectation over $q$. If we choose a variational family with analytic forms for weight means and covariances (e.g. Gaussian with variational parameters $\langle W_{ji} \rangle$ and $\text{Cov}(W_{ji}, W_{lk})$), then the only difficult terms are the moments of $\boldsymbol{h}$:

$$\langle h_j \rangle \propto \int f(\alpha_j) \exp\left[ -\frac{(\alpha_j - \langle a'_j \rangle)^2}{2\Sigma'_{jj}} \right] d\alpha_j, \quad (4)$$

$$\langle h_j h_l \rangle \propto \int f(\alpha_j) f(\alpha_l) \exp\left[ -\frac{1}{2} \begin{pmatrix} \alpha_j - \langle a'_j \rangle \\ \alpha_l - \langle a'_l \rangle \end{pmatrix}^\top \begin{pmatrix} \Sigma'_{jj} & \Sigma'_{jl} \\ \Sigma'_{lj} & \Sigma'_{ll} \end{pmatrix}^{-1} \begin{pmatrix} \alpha_j - \langle a'_j \rangle \\ \alpha_l - \langle a'_l \rangle \end{pmatrix} \right] d\alpha_j \, d\alpha_l, \quad (5)$$

---

[1]We are also required to choose a Gaussian variational approximation for $\boldsymbol{b}$ to preserve the Gaussian distribution of $\boldsymbol{a}$.

| | $A(\mu_1, \mu_2, \rho)$ | $Q(\mu_1, \mu_2, \rho)$ |
|---|---|---|
| Heaviside | $\Phi(\mu_1)\Phi(\mu_2)$ | $-\log(\frac{g_h\rho}{2\pi}) + \frac{\rho}{2g_h\bar{\rho}}\left[\mu_1^2 + \mu_2^2 - \frac{2\rho}{1+\bar{\rho}}\mu_1\mu_2\right] + \mathcal{O}(\mu^4)$ |
| ReLU | $\text{SR}(\mu_1)\text{SR}(\mu_2)$ $+ \rho\Phi(\mu_1)\Phi(\mu_2)$ | $-\log(\frac{g_r}{2\pi}) + \left[\frac{\rho}{2g_r(1+\bar{\rho})}\left(\mu_1^2 + \mu_2^2\right) - \frac{\arcsin\rho - \rho}{g_r\rho}\mu_1\mu_2\right] + \mathcal{O}(\mu^4)$ |

Table 1: Forms for the components of the approximation in equation 6 for Heaviside and ReLU non-linearities. $\Phi$ is the CDF of a standard Gaussian, SR is a "soft ReLU" that we define as $\text{SR}(x) = \phi(x) + x\Phi(x)$ where $\phi$ is a standard Gaussian, $\bar{\rho} = \sqrt{1 - \rho^2}$, $g_h = \arcsin\rho$ and $g_r = g_h + \frac{\rho}{1+\bar{\rho}}$

where we have used the Gaussian form of $\boldsymbol{a}'$ parameterized by mean $\langle\boldsymbol{a}'\rangle$ and covariance $\boldsymbol{\Sigma}'$, and for brevity we have omitted the normalizing constants. Closed form solutions for the integral in equation 4 exist for Heaviside or ReLU choices of non-linearity $f$ (see appendix A). Furthermore, for these non-linearities, the $\langle a_j'\rangle \to \pm\infty$ and $\langle a_l'\rangle \to \pm\infty$ asymptotes of the integral in equation 5 have closed form. Figure 2 shows schematically how these asymptotes can be used as a first approximation for equation 5. This approximation is improved by considering that (by definition) the residual decays to zero far from the origin in the $(\langle a_j'\rangle, \langle a_l'\rangle)$ plane, and so is well modelled by a decaying function $\exp[-Q(\langle a_j'\rangle, \langle a_l'\rangle, \boldsymbol{\Sigma}')]$, where $Q$ is a polynomial in $\langle a'\rangle$ with a dominant positive even term. In practice we truncate $Q$ at the quadratic term, and calculate the polynomial coefficients by matching the moments of the resulting Gaussian with the analytic moments of the residual. Specifically, using dimensionless variables $\mu_i' = \langle a_i'\rangle / \sqrt{\Sigma_{ii}'}$ and $\rho_{ij}' = \Sigma_{ij}'/\sqrt{\Sigma_{ii}'\Sigma_{jj}'}$, this improved approximation takes the form

$$\langle h_j h_l\rangle = S_{jl}'\left\{A(\mu_j', \mu_l', \rho_{jl}') + \exp\left[-Q(\mu_j', \mu_l', \rho_{jl}')\right]\right\}, \tag{6}$$

where the expressions for the dimensionless asymptote $A$ and quadratic $Q$ are given in table table 1 and derived in appendix A.2.1 and A.2.2. The dimensionful scale factor $S_{jl}'$ is 1 for a Heaviside non-linearity or $\Sigma_{jl}'/\rho_{jl}'$ for ReLU. Using equation 6 in equation 3 gives a closed form approximation for the moments of $\boldsymbol{a}$ as a function of moments of $\boldsymbol{a}'$. Since $\boldsymbol{a}$ is approximately normally distributed by the CLT, this is sufficient information to sequentially propagate moments all the way through the network to compute the mean and covariances of $\tilde{q}(\boldsymbol{a}^L)$, our explicit multivariate Gaussian approximation to $q(\boldsymbol{a}^L)$. Any deep learning framework supporting special functions $\arcsin$ and $\Phi$ will immediately support backpropagation through the deterministic expressions we have presented. Below we briefly empirically verify the presented approximation, and in section 3.2 we will show how it is used to compute an approximate log-likelihood and posterior predictive distribution for regression and classification tasks.

### 3.1.1 EMPIRICAL VERIFICATION

**Approximation accuracy** The approximation derived above relies on three assumptions. First, that some form of CLT holds for the hidden units during training where the iid assumption of the classic CLT is not strictly enforced; second, that a quadratic truncation of $Q$ is sufficient[2]; and third that there are only weak correlation between layers so that they can be represented using independent variables in the variational distribution. To provide evidence that these assumptions hold in practice, we train a small ReLU network with two hidden layers each of 128 units

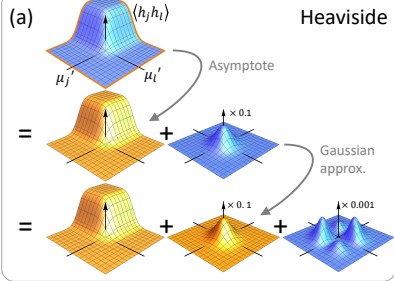
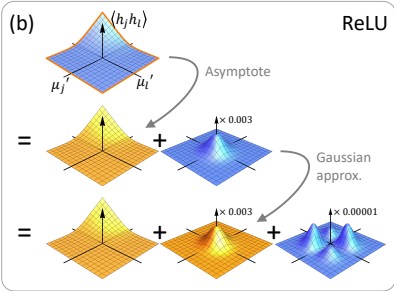

Figure 2: Approximation of $\langle h_j h_l\rangle$ using an asymptote and Gaussian correction for (a) Heaviside and (b) ReLU non-linearities. Yellow functions have closed-forms, and blue indicates residuals. The examples are plotted for $-6 < \mu' < 6$ and $\rho_{jl}' = 0.5$, and the relative magnitude of each correction term is indicated on the vertical axis.

---

[2]Additional Taylor expansion terms can be computed if this assumption fails.

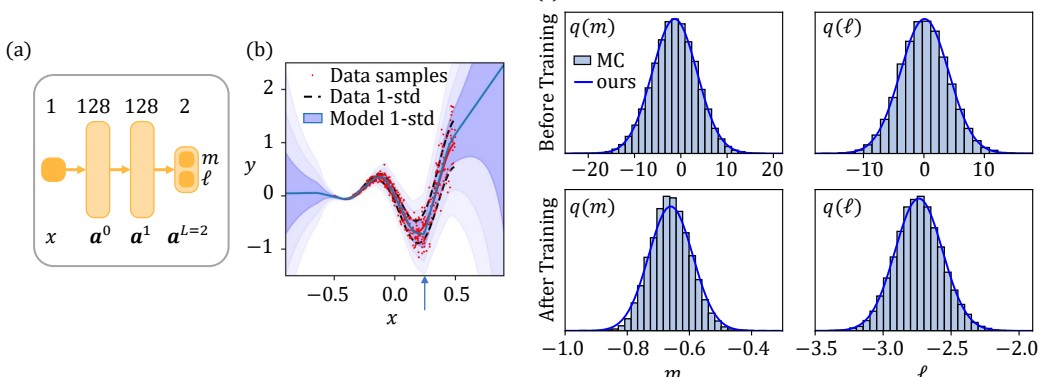

Figure 3: Empirical accuracy of our approximation on toy 1-dimensional data. (a) We train a 2 layer ReLU network to perform heteroscedastic regression on the dataset shown in (b) and obtain the fit shown in blue. (c) The output distributions for the activation units $m$ and $\ell$ evaluated at $x = 0.25$ are in excellent agreement with Monte Carlo (MC) integration with a large number (20k) of samples both before and after training.

to perform 1D heteroscedastic regression on a toy dataset of 500 points drawn from the distribution shown in figure 3(b). Deeper networks and skip connections are considered in appendix C. The training objective is taken from section 4, and the only detail required here is that $\boldsymbol{a}^L$ is a 2-element vector where the elements are labelled as $(m, \ell)$. We use a diagonal Gaussian variational family to represent the weights, but we preserve the full covariance of $\boldsymbol{a}$ during propagation. Using an input $x = 0.25$ (see arrow, Figure 3(b)) we compute the distributions for $m$ and $\ell$ both at the start of training (where we expect the iid assumption to hold) and at convergence (where iid does not necessarily hold). Figure 3(c) shows the comparison between $\boldsymbol{a}^L$ distributions reported by our deterministic approximation and MC evaluation using 20k samples from $q(\boldsymbol{w}; \boldsymbol{\theta})$. This comparison is qualitatively excellent for all cases considered.

**Computational efficiency** In traditional MCVI, propagation of $S$ samples of $d$-dimensional activations through a layer containing a $d \times d$-dimensional transformation requires $\mathcal{O}(Sd^2)$ compute and $\mathcal{O}(Sd)$ memory. Our DVI method approximates the $S \to \infty$ limit, while only demanding $\mathcal{O}(d^3)$ compute and $\mathcal{O}(d^2)$ memory (the additional factor of $d$ arises from manipulation of the quadratically large covariance matrix $\mathrm{Cov}[h_j, h_l]$). Whereas MCVI can always trade compute and memory for accuracy by choosing a small value for $S$, the inherent scaling of DVI with $d$ could potentially limit its practical use for networks with large hidden size. To avoid this limitation, we also consider the case where only the diagonal entries $\mathrm{Cov}(h_j, h_j)$ are computed and stored at each layer. We refer to this method as "diagonal-DVI" (dDVI), and in section 6 we show the surprising result that the strong test performance of DVI is largely retained by dDVI across a range of datasets. Figure 4 shows the time required to propagate activations through a single layer using the MCVI, DVI and dDVI methods

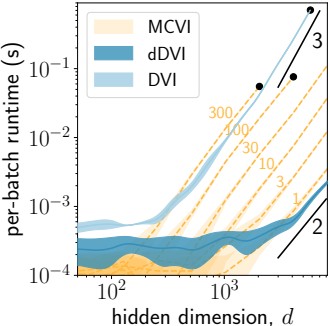

Figure 4: Runtime performance of VI methods. We show the time to propagate a batch of 10 activation vectors through a single $d \times d$ layer. For MCVI we label curves with the number of samples used, and we show quadratic and cubic scaling guides-to-the-eye (black). Black dots indicate where our implementation runs out of memory (16GB).

on a Tesla V100 GPU. As a rough rule of thumb (on this hardware), for layer sizes of practical relevance, we see that absolute DVI runtimes roughly equate to MCVI with $S = 300$ and dDVI runtime equates to $S = 1$.

## 3.2 LOG-LIKELIHOOD EVALUATION

To use the moment propagation procedure derived above for training BNNs, we need to build a function $\mathcal{L}$ that maps final layer activations $\boldsymbol{a}^L$ to the expected log-likelihood term in equation 1 (see figure 1(b)). In appendix B.1 we show the intuitive result that this expected log-likelihood over $q(\boldsymbol{w})$

can be rewritten as an expectation over $\tilde{q}(\boldsymbol{a}^L)$.

$$\mathbb{E}_{\boldsymbol{w}\sim q}\left[\log p(y|\boldsymbol{x},\boldsymbol{w})\right] = \mathbb{E}_{\boldsymbol{a}^L\sim q(\boldsymbol{a}^L)}\left[\log p(y|\boldsymbol{a}^L)\right]. \tag{7}$$

With this form we can derive closed forms for specific tasks; for brevity we focus on the regression case and refer the reader to appendices B.4 and B.5 for the classification case.

**Regression Case**   For simplicity we consider scalar $y$ and a Gaussian noise model parameterized by mean $m(\boldsymbol{x};\boldsymbol{w})$ and heteroscedastic log-variance $\log \sigma_y^2(\boldsymbol{x}) = \ell(\boldsymbol{x};\boldsymbol{w})$. The parameters of this Gaussian are read off as the elements of a 2-dimensional output layer $\boldsymbol{a}^L = (m, \ell)$ so that $p(y|\boldsymbol{a}^L) = \mathcal{N}\left[y|m, e^\ell\right]$. Recall that these parameters themselves are uncertain and the statistics $\langle \boldsymbol{a}^L \rangle$ and $\boldsymbol{\Sigma}^L$ can be computed following section 3.1. Inserting the Gaussian forms for $p(y|\boldsymbol{a}^L)$ and $q(\boldsymbol{a}^L)$ into equation 7 and performing the integral (see appendix B.2) gives a closed form expression for the ELBO reconstruction term:

$$\mathbb{E}_{\boldsymbol{a}^L\sim \tilde{q}(\boldsymbol{a}^L)}\left[\log p(y|\boldsymbol{a}^L)\right] = -\tfrac{1}{2}\left[\log 2\pi + \langle \ell \rangle + \frac{\Sigma_{mm}+(\langle m\rangle - \Sigma_{m\ell}-y)^2}{e^{\langle \ell \rangle - \Sigma_{\ell\ell}/2}}\right]. \tag{8}$$

This heteroscedastic model can be made homoscedastic by setting $\langle \ell \rangle = \Sigma_{\ell\ell} = \Sigma_{m\ell} = 0$. The expression in equation 8 completes the derivations required to implement the closed form approximation to the ELBO reconstruction term for training a network. In addition, we can also compute a closed form approximation to the predictive distribution that is used at test-time to produce predictions that incorporate all parameter uncertainties. By approximating the moments of the posterior predictive and assuming normality (see appendix B.3), we find:

$$p(y) \approx \int p(y|\boldsymbol{a}^L)\,\tilde{q}(\boldsymbol{a}^L)\,d\boldsymbol{a}^L \approx \mathcal{N}\left(y\middle|\langle m\rangle, \Sigma_{mm} + e^{\langle \ell\rangle + \Sigma_{\ell\ell}/2}\right). \tag{9}$$

## 4   EMPIRICAL BAYES FOR VARIATIONAL BNNs

So far, we have described methods for deterministic approximation of the reconstruction term in the ELBO. We now turn to the KL term. For a $d$-dimensional Gaussian prior $p(\boldsymbol{w}) = \mathcal{N}(\boldsymbol{\mu}_{\mathrm{p}}, \boldsymbol{\Sigma}_{\mathrm{p}})$, the KL divergence with the Gaussian variational distribution $q = \mathcal{N}(\boldsymbol{\mu}_{\mathrm{q}}, \boldsymbol{\Sigma}_{\mathrm{q}})$ has closed form:

$$D_{\mathrm{KL}}\left[q||p\right] = \tfrac{1}{2}\left[\log \frac{|\boldsymbol{\Sigma}_{\mathrm{p}}|}{|\boldsymbol{\Sigma}_{\mathrm{q}}|} - d + \mathrm{Tr}\left(\boldsymbol{\Sigma}_{\mathrm{p}}^{-1}\boldsymbol{\Sigma}_{\mathrm{q}}\right) + (\boldsymbol{\mu}_{\mathrm{p}} - \boldsymbol{\mu}_{\mathrm{q}})^\top \boldsymbol{\Sigma}_{\mathrm{p}}^{-1}(\boldsymbol{\mu}_{\mathrm{p}} - \boldsymbol{\mu}_{\mathrm{q}})\right]. \tag{10}$$

However, this requires selection of $(\mu_{\mathrm{p}}, \boldsymbol{\Sigma}_{\mathrm{p}})$ for which there is usually little intuition beyond arguing $\boldsymbol{\mu}_{\mathrm{p}} = \boldsymbol{0}$ by symmetry and choosing $\boldsymbol{\Sigma}_{\mathrm{p}}$ to preserve the expected magnitude of the propagated activations (Glorot & Bengio, 2010; He et al., 2015). In practice, variational Bayes for neural network parameters is sensitive to the choice of prior variance parameters, and we will demonstrate this problem empirically in section 6 (figure 5).

To make variational Bayes robust we parameterize the prior hierarchically, retaining a conditional diagonal Gaussian prior and variational distribution on the weights. The hierarchical prior takes the form $\mathbf{s} \sim p(\mathbf{s}); \boldsymbol{w} \sim p(\boldsymbol{w}|\mathbf{s})$, using an inverse gamma distribution on $\mathbf{s}$ as the conjugate prior to the elements of the diagonal Gaussian variance. We partition the weights into sets $\{\lambda\}$ that typically coincide with the layer partitioning[3], and assign a single element in $\mathbf{s}$ to each set:

$$s_\lambda \sim \text{Inv-Gamma}(\alpha, \beta), \quad w_i^\lambda \sim \mathcal{N}(0, s_\lambda), \tag{11}$$

for shape $\alpha$ and scale $\beta$, and where $w_i^\lambda$ is the $i^{\text{th}}$ weight in set $\lambda$.

Rather than taking the fully Bayesian approach, we adopt an *empirical* Bayes approach (Type-2 MAP), optimizing $s^\lambda$, assuming that the integral is dominated by a contribution from this optimal value $s^\lambda = s_*^\lambda$. We use the data to inform the optimal setting of $s_*^\lambda$ to produce the tightest ELBO:

$$\text{ELBO} = \mathbb{E}_{\boldsymbol{w}\sim q}\left[\log p(y|\boldsymbol{h}^L(\boldsymbol{w}))\right] - \left\{D_{\mathrm{KL}}\left[q(\boldsymbol{w};\boldsymbol{\theta})||p(\boldsymbol{w}|\mathbf{s}_*)p(\mathbf{s}_*)\right]\right\}$$
$$\implies s_*^\lambda = \underset{s^\lambda}{\text{argmin}}\left\{D_{\mathrm{KL}}\left[q(\boldsymbol{w};\boldsymbol{\theta})||p(\boldsymbol{w}^\lambda|s^\lambda)\right] - \log p(s^\lambda)\right\} \tag{12}$$

---

[3]In general, any arbitrary partitioning can be used

Writing out the integral for the KL in equation 12, substituting in the forms of the distributions in equation 11 and differentiating to find the optimum gives

$$s_*^\lambda = \frac{\mathrm{Tr}\left[\boldsymbol{\Sigma}_{\mathrm{q}}^\lambda + \boldsymbol{\mu}_{\mathrm{q}}^\lambda(\boldsymbol{\mu}_{\mathrm{q}}^\lambda)^\top\right] + 2\beta}{\Omega^\lambda + 2\alpha + 2}, \tag{13}$$

where $\Omega^\lambda$ is the number of weights in the set $\lambda$. The influence of the data on the choice of $s_*^\lambda$ is made explicit here through dependence on the learned variational parameters $\boldsymbol{\Sigma}_{\mathrm{q}}$ and $\boldsymbol{\mu}_{\mathrm{q}}$. Using $s_*^\lambda$ to populate the elements of the diagonal prior variance $\boldsymbol{\Sigma}_{\mathrm{p}}$, we can evaluate the KL in equation 10 under the empirical Bayes prior. Optimization of the resulting ELBO then simultaneously tunes the variational distribution and prior.

In the experiments we will demonstrate that the proposed empirical Bayes approach works well; however, it only approximates the full Bayesian solution, and it could fail if we were to allow too many degrees of freedom. To see this, assume we were to use one prior per weight element, and we would also define a hyperprior for each prior mean. Then, adjusting both the prior variance and prior mean using empirical Bayes would always lead to a KL-divergence of zero and the ELBO objective would degenerate into maximum likelihood.

## 5  RELATED WORK

Bayesian neural networks have a rich history. In a 1992 landmark paper David MacKay demonstrated the many potential benefits of a Bayesian approach to neural network learning (MacKay, 1992); in particular, this work contained a convincing demonstration of naturally accounting for model flexibility in the form of the Bayesian *Occam's razor*, facilitating comparison between different models, accurate calibration of predictive uncertainty, and to perform learning robust to overfitting. However, at the time Bayesian inference was achieved only for small and shallow neural networks using a comparatively crude Laplace approximation. Another early review article summarizing advantages and challenges in Bayesian neural network learning is (MacKay, 1995c).

This initial excitement around Bayesian neural networks led to two main methods being developed; First, Hinton & van Camp (1993) and MacKay (1995b) developed the variational Bayes (VB) approach for posterior inference. Whereas Hinton & van Camp (1993) were motivated from a minimum description length (MDL) compression perspective, MacKay (1995b) motivated his equivalent *ensemble learning* method from a statistical physics perspective of variational free energy minimization. Barber & Bishop (1998) extended the methodology for two-layer neural networks to use general multivariate Normal variational distributions. Second, Neal (1993) developed efficient gradient-based Monte Carlo methods in the form of "hybrid Monte Carlo", now known as *Hamiltonian Monte Carlo*, and also raised the question of prior design and limiting behaviour of Bayesian neural networks.

**Rebirth of Bayesian neural networks.** After more than a decade of no further work on Bayesian neural networks Graves (2011) revived the field by using Monte Carlo variational inference (MCVI) to make VB practical and scalable, demonstrating gains in predictive performance on real world tasks.

Since 2015 the VB approach to Bayesian neural networks is mainstream (Blundell et al., 2015); key research drivers since then are the problems of high variance in MCVI and the search for useful variational families. One approach to reduce variance in feedforward networks is the *local reparameterization trick* (Kingma et al., 2015) (see appendix E). To enhance the variational families more complicated distributions such as Matrix Gaussian posteriors (Louizos & Welling, 2016), multiplicative posteriors (Kingma et al., 2015), and hierarchical posteriors (Louizos & Welling, 2017) are used. Both our methods, the deterministic moment approximation and the empirical Bayes estimation, can potentially be extended to these richer families.

**Prior choice.** Choosing priors in Bayesian neural networks remains an open issue. The hierarchical priors for feedforward neural networks that we use have been investigated before by Neal (1993) and MacKay (1995a), the latter proposing a "cheap and cheerful" heuristic, alternating optimization of weights and inverse variance parameters. Barber & Bishop (1998) also used a hierarchical prior and an efficient closed-form factored VB approximation; our approach can be seen as a point estimate to their approach in order to enable use of our closed-form moment approximation. Note that Barber & Bishop (1998) manipulate an expression for $\langle h_j h_l \rangle$ into a one-dimensional integral, whereas our

approach gives closed form approximations for this integral without need for numerical integration. Graves (2011) also used hierarchical Gaussian priors with flat hyperpriors, deriving a closed-form update for the prior mean and variance. Compared to these prior works our approach is rigorous and with sufficient data accurately approximates the Bayesian approach of integrating over the prior parameters.

**Alternative inference procedures.** As an alternative to variational Bayes, *probabilistic backpropagation* (PBP) (Hernández-Lobato & Adams, 2015) applies approximate inference in the form of *assumed density filtering* (ADF) to refine a Gaussian posterior approximation. Like in our work, each update to the approximate posterior requires propagating means and variances of activations through the network. (Hernández-Lobato & Adams, 2015) only consider the diagonal propagation case and homoscedastic regression. Since the original work, PBP has been generalized to classification (Ghosh et al., 2016) and richer posterior families such as the matrix variate Normal posteriors (Sun et al., 2017). Our moment approximation could be used to improve the inference accuracy of PBP, and since we handle minibatches of data rather than processing one data point at a time, our method is more computationally efficient.

**Gaussianity in neural networks.** Our demonstration of Gaussianity of ReLU network activations is also directly relevant to recent work on Gaussian process interpretations of deep neural networks (Matthews et al., 2018; Lee et al., 2017), validating the insight that activations in deep neural networks are closely approximated by Gaussian processes. Two recent works derived deterministic moment approximations for deep neural networks: Bibi et al. (2018), using Price's theorem, derived exact first and second moment expressions for ReLU activations but limit themselves to the case of zero-mean Gaussian activations. Kandemir et al. (2018) also derive closed-form solutions to the ELBO for the case of diagonal Gaussian variational families. However, their approach is limited to linear layers without bias.

**Markov chain Monte Carlo approaches.** Another rich class of approximate inference methods for Bayesian neural networks are stochastic gradient Markov chain Monte Carlo (SG-MCMC) methods. These methods allow for approximate posterior parameter inference using unbiased log-likelihood estimates. Stochastic gradient Langevin dynamics (SGLD) was the first method in this class (Welling & Teh, 2011). SGLD is particularly simple and efficient to implement, but recent methods increase efficiency in the case of correlated posteriors by estimating the Fisher information matrix (Ahn et al., 2012) and extend Hamiltonian Monte Carlo to the stochastic gradient case (Chen et al., 2014). A *complete* characterization of SG-MCMC methods is given by (Ma et al., 2015; Gong et al., 2018). However, despite this progress, important theoretical questions regarding approximation guarantees for practical computational budgets remain (Nagapetyan et al., 2017). Moreover, while SG-MCMC methods work robustly in practice, they remain computationally inefficient, especially because evaluation of the posterior predictive requires evaluating an ensemble of models.

**Wild approximations.** The above methods are principled but often require sophisticated implementations; recently, a few methods aim to provide "cheap" approximations to the Bayes posterior. Dropout has been interpreted by Gal & Ghahramani (2016) to approximately correspond to variational inference. Likewise, *Bootstrap posteriors* (Lakshminarayanan et al., 2017; Fushiki et al., 2005; Harris, 1989) have been proposed as a general, robust, and accurate method for posterior inference. However, obtaining a bootstrap posterior ensemble of size $k$ is computationally intense at $k$ times the computation of training a single model.

## 6 EXPERIMENTS

We implement[4] deterministic variational inference (DVI) as described above to train small ReLU networks on UCI regression datasets (Dheeru & Karra Taniskidou, 2017). The experiments address the claims that our methods for eliminating gradient variance and automatic tuning of the prior improve the performance of the final trained model. In Appendix D we present extended results to demonstrate that our method is competitive against a variety of models and inference schemes.

---

[4]Our implementation in TensorFlow is available at `https://github.com/Microsoft/deterministic-variational-inference`

| Dataset | $|\mathcal{D}|$ | $d_x$ | DVI | dDVI | MCVI | hoDVI |
|---------|------|-------|-----|------|------|-------|
| bost | 506 | 13 | $\mathbf{-2.41 \pm 0.02}$ | $-2.42 \pm 0.02$ | $-2.46 \pm 0.02$ | $-2.58 \pm 0.04$ |
| conc | 1030 | 8 | $\mathbf{-3.06 \pm 0.01}$ | $-3.07 \pm 0.02$ | $-3.07 \pm 0.01$ | $-3.23 \pm 0.01$ |
| ener | 768 | 8 | $\mathbf{-1.01 \pm 0.06}$ | $-1.06 \pm 0.06$ | $-1.03 \pm 0.04$ | $-2.09 \pm 0.06$ |
| kin8 | 8192 | 8 | $1.13 \pm 0.00$ | $1.13 \pm 0.00$ | $\mathbf{1.14 \pm 0.00}$ | $1.01 \pm 0.01$ |
| nava | 11934 | 16 | $\mathbf{6.29 \pm 0.04}$ | $6.22 \pm 0.06$ | $5.94 \pm 0.05$ | $5.84 \pm 0.06$ |
| powe | 9568 | 4 | $\mathbf{-2.80 \pm 0.00}$ | $\mathbf{-2.80 \pm 0.00}$ | $\mathbf{-2.80 \pm 0.00}$ | $-2.82 \pm 0.00$ |
| prot | 45730 | 9 | $-2.85 \pm 0.01$ | $\mathbf{-2.84 \pm 0.01}$ | $-2.87 \pm 0.01$ | $-2.94 \pm 0.00$ |
| wine | 1588 | 11 | $\mathbf{-0.90 \pm 0.01}$ | $-0.91 \pm 0.02$ | $-0.92 \pm 0.01$ | $-0.96 \pm 0.01$ |
| yach | 308 | 6 | $\mathbf{-0.47 \pm 0.03}$ | $\mathbf{-0.47 \pm 0.03}$ | $-0.68 \pm 0.03$ | $-1.41 \pm 0.03$ |

Table 2: Average test log-likelihood on UCI datasets. $|\mathcal{D}|$ is the dataset size, and $d_x$ is the input dimension.

**Deterministic vs. Stochastic**  We compare DVI with MCVI from equation 2 with $S = 10$ samples (we consider vanilla MCVI and discuss the local reparameterization trick in appendix E). The same model is used for each inference method: a single hidden layer of 50 units for each dataset considered, extending this to 100 units in the special case of the larger protein structure dataset, prot. Although neither DVI nor MCVI is limited to a particular choice of variational family $q(\boldsymbol{w}; \boldsymbol{\theta})$, we use a factorized Gaussian family (i.e. a diagonal $\mathrm{Cov}(W_{ji}, W_{lk})$). Factorization reduces the computational complexity of terms involving $\mathrm{Cov}(W_{ji}, W_{lk})$ in DVI[5] from $\mathcal{O}(N^2)$ to $\mathcal{O}(N)$, where $N$ is the number of elements in $\boldsymbol{W}$ (see appendix A.1). Additionally, both methods use the same EB prior from equation 13 with a broad inverse Gamma hyperprior ($\alpha = 1$, $\beta = 10$) and an independent $s_\lambda$ for each linear transformation. Each dataset is split into random training and test sets with 90% and 10% of the data respectively. This splitting process is repeated 20 times and the average test performance of each method at convergence is reported in table 2 (see also learning curves in appendix F). We see that DVI consistently outperforms MCVI, by up to 0.35 nats per data point on some datasets. The computationally efficient diagonal-DVI (dDVI) surprisingly retains much of this performance. By default we use the heteroscedastic model, and we observe that this uniformly delivers better results than a homoscedastic model (hoDVI; rightmost column in table 2) on these datasets with no overfitting issues[6].

**Empirical Bayes**  In Figure 5 we compare the performance of networks trained with manual tuning of a fixed Gaussian prior to networks trained with the automatic EB tuning. We find that the EB method consistently finds priors that produce models with competitive or significantly improved test log-likelihood relative to the best manual setting. Since this observation holds across all datasets considered, we say that our method is "robust". Note that the EB method can outperform manual tuning because it automatically finds different prior variances for each weight matrix, whereas in the manual tuning case we search over a single hyperparameter control-

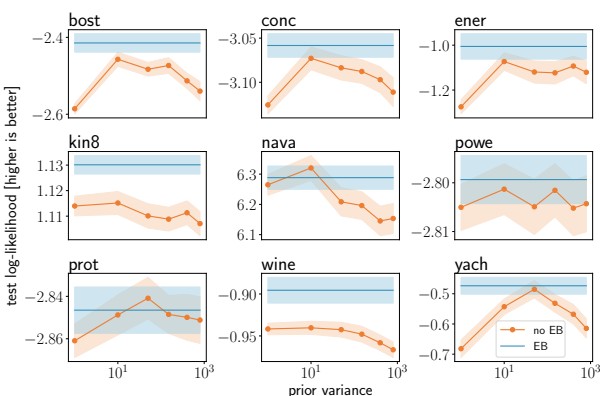

Figure 5: Comparison of converged test log-likelihood with a manually tuned prior variance (orange) or empirical Bayes (blue).

ling all prior variances. An additional ablation study showing the relative contribution of our deterministic approach and the EB prior are shown in appendix D.1.

---

[5]For MCVI with full rank covariance, Cholesky decomposition required for sampling is $\mathcal{O}(N^3)$.

[6]Note that this result is non-trivial because heteroscedastic models are more complex and could result in poorer approximate inference leading to worse test performance

## 7 CONCLUSION

We introduced two innovations to make variational inference for neural networks more robust: 1. an effective deterministic approximation to the moments of activations of a neural networks; and 2. a simple empirical Bayes hyperparameter update. We demonstrate that together these innovations make variational Bayes a competitive method for Bayesian inference in neural heteroscedastic regression models.

Bayesian neural networks have been shown to substantially improve upon standard networks in these settings where calibrated predictive uncertainty estimates, sequential decision making, or continual learning without catastrophic forgetting are required (see e.g. Oliveira et al. (2016); Gal et al. (2017); Nguyen et al. (2018)). In future work, the new innovations proposed in this paper can be applied to these areas. In the sequential decision making and continual learning applications, approximate Bayesian inference must be run as an inner loop of a larger algorithm. This requires a robust and automated version of BNN training: this is precisely where we believe the innovations in this paper will have large impact since they pave the way to automated and robust deployment of BBNs that do not involve an expert in-the-loop.

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

## APPENDIX

## A   MOMENTS OF THE ACTIVATION VARIABLES $a^\ell$

Under assumption of independence of $\boldsymbol{h}$, $\boldsymbol{W}$ and $\boldsymbol{b}$, we can write:

$$\langle a_i \rangle = \langle h_j W_{ji} \rangle + \langle b_i \rangle = \langle h_j \rangle \langle W_{ji} \rangle + \langle b_i \rangle \tag{14}$$

$$
\begin{aligned}
\mathrm{Cov}(a_i, a_k) &= \mathrm{Cov}(h_j W_{ji}, h_l W_{lk}) + \mathrm{Cov}(b_i, b_k) \\
&= \langle h_j W_{ji} \, h_l W_{lk} \rangle - \langle h_j W_{ji} \rangle \langle h_l W_{lk} \rangle + \mathrm{Cov}(b_i, b_k) \\
&= \langle h_j h_l \rangle \langle W_{ji} W_{lk} \rangle - \langle h_j \rangle \langle h_l \rangle \langle W_{ji} \rangle \langle W_{lk} \rangle + \mathrm{Cov}(b_i, b_k) \\
&= \langle h_j h_l \rangle \left[ \mathrm{Cov}(W_{ji} W_{lk}) + \langle W_{ji} \rangle \langle W_{lk} \rangle \right] \\
&\quad - \langle h_j \rangle \langle h_l \rangle \langle W_{ji} \rangle \langle W_{lk} \rangle + \mathrm{Cov}(b_i, b_k) \\
&= \langle h_j h_l \rangle \mathrm{Cov}(W_{ji}, W_{lk}) + \langle W_{ji} \rangle \mathrm{Cov}(h_j, h_l) \langle W_{lk} \rangle + \mathrm{Cov}(b_i, b_k),
\end{aligned} \tag{15}
$$

which is seen in the main text as equation 3. For Heaviside and ReLU activation functions, closed forms exist for $\langle h_j \rangle$ in equation 14:

Heaviside   $\langle h_j \rangle = \frac{1}{\sqrt{2\pi \Sigma'_{jj}}} \int_0^\infty e^{-\frac{1}{2\Sigma'_{jj}}\left(\alpha_j - \langle a'_j \rangle\right)^2} \, \mathrm{d}\alpha_j = \Phi\left(\mu'_j\right)$

ReLU       $\langle h_j \rangle = \frac{1}{\sqrt{2\pi \Sigma'_{jj}}} \int_0^\infty \alpha_j e^{-\frac{1}{2\Sigma'_{jj}}\left(\alpha_j - \langle a'_j \rangle\right)^2} \, \mathrm{d}\alpha_j = \sqrt{\Sigma'_{jj}}\,\mathrm{SR}(\mu'_j),$

where $\mathrm{SR}(x) := \phi(x) + x\Phi(x)$ is a "soft ReLU", $\phi$ and $\Phi$ represent the standard Gaussian PDF and CDF, and we have introduced the dimensionless variables $\mu'_j = \langle a'_j \rangle / \sqrt{\Sigma'_{jj}}$. These results are is sufficient to evaluate equation 14, so in the following sections we turn to each term from equation 15.

### A.1   EVALUATION OF TERM 1: $\langle h_j h_l \rangle \, \mathrm{Cov}(W_{ji}, W_{lk})$

In the general case, we can use the results from section A.2 to evaluate off-diagonal $\langle h_j h_l \rangle$. However, in our experiments we always consider the the special case where $\mathrm{Cov}(W_{ji}, W_{lk})$ is diagonal. In this case we can write the first term in equation 15 as (reintroducing the explicit summation):

$$
\begin{aligned}
\sum_{jl} \langle h_j h_l \rangle \, \mathrm{Cov}(W_{ji}, W_{lk}) &= \sum_{jl} \langle h_j h_l \rangle \, \delta_{jl} \delta_{ik} \mathrm{Var}(W_{ji}) \\
&= \delta_{ik} \sum_j \langle z_j z_j \rangle \, \mathrm{Var}(W_{ji}) \\
&= \mathrm{diag}\left[\mathbf{v}\mathrm{Var}(\boldsymbol{W})\right]
\end{aligned}
$$

i.e. this term is a diagonal matrix with the diagonal given by the left product of the vector $v_j = \langle h_j h_j \rangle$ with the matrix $\mathrm{Var}(W_{ki})$. Note that $\langle h_j h_j \rangle$ can be evaluated analytically for Heaviside and ReLU activation functions:

Heaviside   $\langle h_j h_j \rangle = \frac{1}{\sqrt{2\pi \Sigma'_{jj}}} \int_0^\infty e^{-\frac{1}{2\Sigma'_{jj}}\left(\alpha_j - \langle a'_j \rangle\right)^2} \, \mathrm{d}\alpha_j = \Phi\left(\mu'_j\right)$

ReLU       $\langle h_j h_j \rangle = \frac{1}{\sqrt{2\pi \Sigma'_{jj}}} \int_0^\infty \alpha_j^2 e^{-\frac{1}{2\Sigma'_{jj}}\left(\alpha_j - \langle a'_j \rangle\right)^2} \, \mathrm{d}\alpha_j = \Sigma'_{jj}\left[\mu'_j \phi(\mu'_j) + (1 + \mu'^2_j)\Phi(\mu'_j)\right]$

### A.2   EVALUATION OF TERM 2: $\langle W_{ji} \rangle \, \mathrm{Cov}(h_j, h_l) \langle W_{lk} \rangle$

Evaluation of $\mathrm{Cov}(h_j, h_l)$ requires an expression for $\langle h_j h_l \rangle$. From equation 5, we write:

$$\langle h_j h_l \rangle \propto \int f(\alpha_j) f(\alpha_l) \exp\left[-\tfrac{1}{2}P(\alpha_j, \alpha_l; \boldsymbol{a}', \boldsymbol{\Sigma}')\right] \, \mathrm{d}\alpha_j \mathrm{d}\alpha_l, \tag{16}$$

where $P$ is the quadratic form:

$$P(\alpha_j, \alpha_l; \boldsymbol{a}', \boldsymbol{\Sigma}') = \begin{pmatrix} \alpha_j - \langle a'_j \rangle \\ \alpha_l - \langle a'_l \rangle \end{pmatrix}^\top \begin{pmatrix} \Sigma'_{jj} & \Sigma'_{jl} \\ \Sigma'_{lj} & \Sigma'_{ll} \end{pmatrix}^{-1} \begin{pmatrix} \alpha_j - \langle a'_j \rangle \\ \alpha_l - \langle a'_l \rangle \end{pmatrix}$$

$$= \begin{pmatrix} \eta_j - \mu'_j \\ \eta_l - \mu'_l \end{pmatrix}^\top \begin{pmatrix} 1 & \rho'_{jl} \\ \rho'_{lj} & 1 \end{pmatrix}^{-1} \begin{pmatrix} \eta_j - \mu'_j \\ \eta_l - \mu'_l \end{pmatrix}.$$

Here we have introduced further dimensionless variables $\eta_j = \alpha_j/\sqrt{\Sigma'_{jj}}$, $\eta_l = \alpha_l/\sqrt{\Sigma'_{ll}}$ and $\rho'_{jl} = \Sigma'_{jl}/\sqrt{\Sigma'_{jj}\Sigma'_{ll}}$. We can then rewrite equation 16 in terms of a dimensionless integral $I$ using a scale factor $S'_{jl}$ that is 1 for the Heaviside non-linearity or $\Sigma'_{jl}/\rho'_{jl}$ for ReLU:

$$\langle h_j h_l \rangle = S'_{jl} I(\mu'_j, \mu'_l, \rho'_{jl}) ; \quad I = \frac{1}{Z} \int f(\eta_j) f(\eta_l) \exp\left[-\tfrac{1}{2} P(\eta_j, \eta_l; \boldsymbol{\mu}', \boldsymbol{\rho}')\right] \, d\eta_j d\eta_l.$$

The normalization constant, $Z$, is evaluated by integrating over $e^{-P/2}$ and is explicitly written as $Z = 2\pi\bar{\rho}'_{jl}$, where $\bar{\rho}'_{jl} = \sqrt{1 - \rho'^2_{jl}}$. Now, following equation 6, we have the task to write $I$ as an asymptote $A$ plus a decaying correction $e^{-Q}$. To evaluate $A$ and $Q$, we have to insert the explicit form of the non-linearity $f$, which we do for Heaviside and ReLU functions in the next sections.

### A.2.1 HEAVISIDE NON-LINEARITY

For the Heaviside activation, we can represent the integral $I$ as the shaded area under the Gaussian in the upper-left quadrant shown below. In general, this integral does not have a closed form. However, for $\mu'_j \to \infty$, vanishing weight appears under the Gaussian in the upper-right quadrant, so we can write down the asymptote of the integral in this limit:

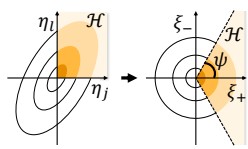

$$\lim_{\mu_j \to \infty} I = \frac{1}{Z} \int_{\eta_j = -\infty}^{\infty} \int_{\eta_l = 0}^{\infty} \exp\left[-\tfrac{1}{2} P(\eta_j, \eta_l; \boldsymbol{\mu}', \rho'_{jl})\right] \, d\eta_j d\eta_l = \Phi(\mu'_l)$$

Here we performed the integral by noticing that the outer integral over $\eta_j$ marginalizes out $\eta_j$ from the bivariate Gaussian, leaving the inner integral as the definition of the Gaussian CDF. By symmetry, we also have $\lim_{\mu'_l \to \infty} I = \Phi(\mu'_j)$ and $\lim_{\mu'_{j,l} \to -\infty} I = 0$. We can then write down the following symmetrized form that satisfies all the limits required to qualify as an asymptote:

$$A = \Phi(\mu'_j)\Phi(\mu'_l)$$

To compute the correction factor we evaluate the derivatives of $(I - A)$ at the origin up to second order to match the moments of $e^{-Q}$ for quadratic $Q$. Description of this process is found below

**Zeroth derivative** At the origin $\mu_j = \mu_l = 0$, we can diagonalize the quadratic form $P$:

$$P(\eta_j, \eta_l; \boldsymbol{0}, \rho'_{jl}) = \tfrac{1}{2\bar{\rho}'^2_{jl}} \left(\eta_j^2 - 2\rho'_{jl}\eta_j\eta_l + \eta_l^2\right) = \tfrac{1}{4\bar{\rho}'^2_{jl}} \left(\xi_+^2 + \xi_-^2\right),$$

where $\xi_\pm = \sqrt{1 \mp \rho}(\eta_1 \pm \eta_2)$. Performing this change of variables in the integral gives:

$$I = \frac{1}{4\pi\bar{\rho}'^2_{jl}} \int_{\mathcal{H}} \exp\left[\tfrac{1}{4\bar{\rho}'^2_{jl}} \left(\xi_+^2 + \xi_-^2\right)\right] \, d\xi_+ d\xi_- = \psi/\pi$$

where we integrated in polar coordinates over the region $\mathcal{H}$ in which the Heaviside function is non-zero. The angle $\psi$ can be found from the coordinate transform between $\eta$ and $\xi$ as[7]:

$$\psi = \arctan\sqrt{\frac{1+\rho'_{jl}}{1-\rho'_{jl}}} = \frac{\pi}{2} - \frac{1}{2}\arccos\rho'_{jl}.$$

Since $A|_{\boldsymbol{\mu}'=\boldsymbol{0}} = \Phi(0)\Phi(0) = 1/4$, we can evaluate:

$$(I - A)|_{\boldsymbol{\mu}'=\boldsymbol{0}} = \frac{\pi}{2} - \frac{1}{2}\arccos\rho'_{jl} - \frac{1}{4}$$

$$= \frac{1}{2\pi}\arcsin\rho_{jl}$$

---

[7]Here we use the identity $\cos(2\arctan x) = \cos^2\arctan x - \sin^2\arctan x = \frac{1-x^2}{1+x^2}$

**First derivative**   Performing a change of variables $x_i = \eta_i - \mu_i'$, we can write $I$ as:

$$I = \frac{1}{Z} \int H(x_j + \mu_j') H(x_l + \mu_l') \exp\left[-\tfrac{1}{2}P(x_j, x_l; \mathbf{0}, \rho_{jl}')\right] \, \mathrm{d}x_j \mathrm{d}x_l$$

where $H$ is the Heaviside function. Now, using $\partial_x H(x) = \delta(x)$, we have:

$$\left.\frac{\partial}{\partial \mu_j'}\right|_{\boldsymbol{\mu}'=\mathbf{0}} I = \frac{1}{Z} \int \delta(x_j) H(x_l) \exp\left[-\tfrac{1}{2}P(x_j, x_l; \mathbf{0}, \rho_{jl}')\right] \, \mathrm{d}x_j \mathrm{d}x_l = \frac{1}{2\sqrt{2\pi}}. \tag{17}$$

In addition, using $\partial_x \Phi(x) = \phi(x)$, we have:

$$\left.\frac{\partial}{\partial \mu_j'}\right|_{\boldsymbol{\mu}'=\mathbf{0}} A = \frac{1}{2\sqrt{2\pi}} \qquad \Longrightarrow \qquad \left.\frac{\partial}{\partial \mu_j'}\right|_{\boldsymbol{\mu}'=\mathbf{0}} (I - A) = 0.$$

By symmetry $(I - A)$ also has zero gradient with respect to $\mu_l'$ at the origin. Therefore $Q$ has no linear term in $\boldsymbol{\mu}'$.

**Second derivative**   Taking another derivative in equation 17 gives:

$$\left.\frac{\partial^2}{\partial \mu_j'^2}\right|_{\boldsymbol{\mu}'=\mathbf{0}} I = -\frac{1}{Z} \int \delta(x_j) H(x_l) \frac{\rho_{jl}'}{\bar{\rho}_{jl}'^2} x_l \exp\left[-\tfrac{1}{2}P(x_j, x_l; \mathbf{0}, \rho_{jl}')\right] \, \mathrm{d}x_j \mathrm{d}x_l = -\frac{\rho_{jl}'}{2\pi \bar{\rho}_{jl}'}.$$

where we used the identity $\int f(x) \partial_x \delta(x) \mathrm{d}x = -\int \delta(x) \partial_x f(x) \mathrm{d}x$, which holds for arbitrary $f$. In addition, we have:

$$\left.\frac{\partial^2}{\partial \mu_j'^2}\right|_{\boldsymbol{\mu}'=\mathbf{0}} A = 0 \qquad \Longrightarrow \qquad \left.\frac{\partial^2}{\partial \mu_j'^2}\right|_{\boldsymbol{\mu}'=\mathbf{0}} (I - A) = -\frac{\rho_{jl}'}{2\pi \bar{\rho}_{jl}'}.$$

and the same result holds for the second derivative w.r.t. $\mu_l'$. To complete the Hessian, it is a simple extension of previous results to show that:

$$\left.\frac{\partial^2}{\partial \mu_j' \partial \mu_l'}\right|_{\boldsymbol{\mu}'=\mathbf{0}} (I - A) = \frac{1 - \bar{\rho}_{jl}'}{2\pi \bar{\rho}_{jl}'}.$$

Now that we have obtained derivatives of the residual $(I - A)$ up to second order we propose a correction factor of the form $e^{-Q}$ where $Q$ is truncated at quadratic terms:

$$Q = -\log \frac{\alpha}{2\pi} + \beta\left(\mu_j'^2 + \mu_l'^2\right) + \gamma \mu_j'^2 \mu_l'^2.$$

We then find the coefficients $\{\alpha, \beta, \gamma\}$ by matching ($\overset{!}{=}$) derivatives at $\boldsymbol{\mu} = \mathbf{0}$:

$$\left. e^{-Q} \right|_{\boldsymbol{\mu}=\mathbf{0}} = \frac{\alpha}{2\pi} \qquad \overset{!}{=} \frac{\arcsin \rho_{jl}'}{2\pi} \implies \alpha = \arcsin \rho_{jl}'$$

$$\left. \frac{\partial}{\partial \mu_i'} e^{-Q} \right|_{\boldsymbol{\mu}=\mathbf{0}} = 0 \qquad \overset{!}{=} 0$$

$$\left. \frac{\partial^2}{\partial \mu_i'^2} e^{-Q} \right|_{\boldsymbol{\mu}=\mathbf{0}} = -2\beta \frac{\alpha}{2\pi} \qquad \overset{!}{=} -\frac{\rho_{jl}'}{2\pi \bar{\rho}_{jl}'} \implies \beta = \frac{\rho_{jl}'}{2\alpha \bar{\rho}_{jl}'}$$

$$\left. \frac{\partial^2}{\partial \mu_j' \partial \mu_l'} e^{-Q} \right|_{\boldsymbol{\mu}=\mathbf{0}} = \gamma \frac{\alpha}{2\pi} \qquad \overset{!}{=} \frac{1 - \bar{\rho}_{jl}'}{2\pi \bar{\rho}_{jl}'} \implies \gamma = \frac{1 - \bar{\rho}_{jl}'}{\alpha \bar{\rho}_{jl}'}$$

This yields the expression seen in table 1 of the main text.

### A.2.2   ReLU non-linearity

As in the Heaviside case, we begin by computing the asymptote of $I$ by inspecting the limit as $\mu_j' \to \infty$:

$$\lim_{\mu_j \to \infty} I = \frac{1}{Z} \int_{\eta_j = -\infty}^{\infty} \int_{\eta_l=0}^{\infty} \eta_j \eta_l \exp\left[-\tfrac{1}{2}P(\eta_j, \eta_l; \boldsymbol{\mu}', \rho_{jl}')\right] \, \mathrm{d}\eta_j \mathrm{d}\eta_l$$

$$= \frac{1}{\sqrt{2\pi}} \int_{\eta_l=0}^{\infty} \mu_j' \eta_l e^{-\frac{1}{2}(\eta_l - \mu_l')^2} \mathrm{d}\eta_l + \frac{\rho_{jl}'}{\sqrt{2\pi}} \int_{\eta_l=0}^{\infty} \eta_l(\eta_l - \mu_l) e^{-\frac{1}{2}(\eta_l - \mu_l')^2} \mathrm{d}\eta_l$$

$$= \mu_j' \, \mathrm{SR}(\mu_l') + \rho_{jl}' \Phi(\mu_l') \tag{18}$$

Now, we construct a full 2-dimensional asymptote by symmetrizing equation 18 (using properties $\mathrm{SR}(x) \to x$ and $\Phi(x) \to 1$ as $x \to \infty$ to check that the correct limits are preserved after symmetrizing):

$$A = \mathrm{SR}(\mu_j')\mathrm{SR}(\mu_l') + \rho_{jl}'\Phi(\mu_j')\Phi(\mu_l')$$

Next we compute the correction factor $e^{-Q}$. The details of this procedure closely follow those for the Heaviside non-linearity of the previous section, so we omit them here (and in practice we use `Mathematica` to perform the intermediate calculations). The final result is presented in table 1 of the main text.

## B   LOG-LIKELIHOOD AND POSTERIOR PREDICTIVE COMPUTATION

Here we give derivations of expressions quoted in section 3.2. In section B.1 we justify the intuitive result that expectation of the ELBO reconstruction term over $q(\boldsymbol{w}; \boldsymbol{\theta})$ can be re-written as an expectation over $\tilde{q}(\boldsymbol{a}^L)$. We then derive expected log-likelihoods and posterior predictive distributions for the cases of univariate Gaussian regression and classification. The latter sections are arranged as follows:

|  | Regression | Classification |
|---:|:---:|:---:|
| Log-likelihood | section B.2 | section B.4 |
| Posterior predictive | section B.3 | section B.5 |

### B.1   LOG-LIKELIHOODS: FROM $\mathbb{E}_w$ TO $\mathbb{E}_{a^L}$

We begin by rewriting the reconstruction term for data point $(\mathbf{x}, y)$ in terms of $\boldsymbol{a}^L$:

$$\mathbb{E}_{\boldsymbol{w} \sim q}\left[\log p(y|\boldsymbol{w})\right] = \int q(\boldsymbol{w}) \log p(y|\boldsymbol{w}) \, \mathrm{d}\boldsymbol{w} = \int q(\boldsymbol{a}^L)q(\boldsymbol{w}|\boldsymbol{a}^L) \log p(y|\boldsymbol{w}) \, \mathrm{d}\boldsymbol{w} \, \mathrm{d}\boldsymbol{a}^L$$

where we have suppressed explicit conditioning on $\boldsymbol{x}$ for brevity. Our goal now is to perform the integral over $\boldsymbol{w}$, leaving the expectation in terms of $\boldsymbol{a}^L$ only, thus allowing it to be evaluated using the approximation $\tilde{q}(\boldsymbol{a}^L)$ from section 3.1.

To eliminate $\boldsymbol{w}$, consider the case where the output of the model is a distribution $p(y|\boldsymbol{a}^L)$ that is a parameter-free transformation of $\boldsymbol{a}^L$ (e.g. $\boldsymbol{a}^L$ are logits of a softmax distribution for classification or the moments of a Gaussian for regression). Since the model output is conditioned only on $\boldsymbol{a}^L$, we must have $p(y|\boldsymbol{w}) = p(y|\boldsymbol{a}^L)$ for all configurations $\boldsymbol{w}$ that satisfy the deterministic transformation $\boldsymbol{a}^L = \mathcal{M}(\mathbf{x}; \boldsymbol{w})$, where $\mathcal{M}$ is the neural network (i.e $p(y|\boldsymbol{w}) = p(y|\boldsymbol{a}^L)$ for all $\boldsymbol{w}$ where $q(\boldsymbol{w}|\boldsymbol{a}^L)$ is non-zero). This allows us to write:

$$\int q(\boldsymbol{w}|\boldsymbol{a}^L) \log p(y|\mathbf{x}, \boldsymbol{w}) \, \mathrm{d}\boldsymbol{w} = \log p(y|\boldsymbol{a}^L) \int q(\boldsymbol{w}|\boldsymbol{a}^L) \, \mathrm{d}\boldsymbol{w} = \log p(y|\boldsymbol{a}^L),$$

so the reconstruction term becomes:

$$\mathbb{E}_{\boldsymbol{w} \sim q}\left[\log p(y|\mathbf{x}, \boldsymbol{w})\right] = \int q(\boldsymbol{a}^L) \log p(y|\boldsymbol{a}^L)\mathrm{d}\boldsymbol{a}^L = \mathbb{E}_{\boldsymbol{a}^L \sim q(\boldsymbol{a}^L)}\left[\log p(y|\boldsymbol{a}^L)\right].$$

This establishes the equivalence given in equation 7 in the main text. Since we are using an approximation to $q$, we will actually compute $\mathbb{E}_{\boldsymbol{a}^L \sim \tilde{q}(\boldsymbol{a}^L)}\left[\log p(y|\boldsymbol{a}^L)\right]$.

### B.2   UNIVARIATE REGRESSION: LOG-LIKELIHOOD

Here we give a derivation of equation 8 from the main text. Throughout this section we label the 2 elements of the final activation vector as $\boldsymbol{a}^L = (m, \ell)$. We first insert the Gaussian form for $p(y|\boldsymbol{a}^L) \sim \mathcal{N}\left[m, e^\ell\right]$ into the log-likelihood expression:

$$\mathbb{E}_{\boldsymbol{a}^L \sim \tilde{q}(\boldsymbol{a}^L)}\left[\log p(y|\boldsymbol{a}^L)\right] = -\tfrac{1}{2}\mathbb{E}_{\boldsymbol{a}^L \sim \tilde{q}(\boldsymbol{a}^L)}\left[\log\left(2\pi \exp(\ell)\right) + \exp(-\ell)(y-m)^2\right]$$
$$= -\tfrac{1}{2}\log 2\pi - \tfrac{1}{2}\left\langle \ell \right\rangle - \mathbb{E}_{\boldsymbol{a}^L \sim \tilde{q}(\boldsymbol{a}^L)}\left[\exp(-\ell)(y-m)^2\right]. \qquad (19)$$

Now we use the Gaussian form of $\tilde{q}(\boldsymbol{a}^L)$:

$$\tilde{q}(\boldsymbol{a}^L) \propto \exp\left[-\tfrac{1}{2}\boldsymbol{X}^\top(\boldsymbol{\Sigma}^L)^{-1}\boldsymbol{X}\right] \;; \quad \boldsymbol{X} = \begin{pmatrix} m - \langle m \rangle \\ \ell - \langle \ell \rangle \end{pmatrix}.$$

and note that

$$\int \exp(-\ell)\exp\left[-\tfrac{1}{2}\boldsymbol{X}^\top(\boldsymbol{\Sigma}^L)^{-1}\boldsymbol{X}\right] \, \mathrm{d}m \, \mathrm{d}\ell$$

$$= \exp\left(-\langle \ell \rangle\right)\int \exp\left[-\tfrac{1}{2}\boldsymbol{X}^\top(\boldsymbol{\Sigma}^L)^{-1}\boldsymbol{X} - (\ell - \langle \ell \rangle)\right] \, \mathrm{d}m \, \mathrm{d}\ell$$

$$= \exp\left(-\langle \ell \rangle\right)\int \exp\left[-\tfrac{1}{2}\boldsymbol{X}^\top(\boldsymbol{\Sigma}^L)^{-1}\boldsymbol{X} - \boldsymbol{e}_\ell^\top \boldsymbol{X}\right] \, \mathrm{d}m \, \mathrm{d}\ell$$

$$= \exp\left(\tfrac{\Sigma_{\ell\ell}}{2} - \langle \ell \rangle\right)\int \exp\left[-\tfrac{1}{2}(\boldsymbol{X}^\top + \boldsymbol{e}_\ell^\top\boldsymbol{\Sigma}^L)(\boldsymbol{\Sigma}^L)^{-1}(\boldsymbol{X} + \boldsymbol{e}_\ell\boldsymbol{\Sigma}^L)\right] \, \mathrm{d}m \, \mathrm{d}\ell, \qquad (20)$$

where $\boldsymbol{e}_\ell^\top = (0,1)$ is the unit vector in the $\ell$ coordinate, and we completed the square to obtain the final line. Inserting equation 20 into equation 19 and marginalizing out the $\ell$ coordinate gives:

$$\mathbb{E}_{\boldsymbol{a}^L \sim \tilde{q}(\boldsymbol{a}^L)}\left[\log p(y|\boldsymbol{a}^L)\right] = -\tfrac{1}{2}\left[\log 2\pi + \langle \ell \rangle + \frac{e^{\Sigma_{\ell\ell}/2 - \langle \ell \rangle}}{\sqrt{2\pi\Sigma_{mm}}}\int (y-m)^2 \exp\left(-\frac{[m - (\langle m \rangle - \Sigma_{m\ell})]^2}{2\Sigma_{mm}}\right) \, \mathrm{d}m\right].$$

Finally, performing the integral over $m$ gives the result seen in equation 8.

### B.3 Univariate Regression: Posterior Predictive Distribution

Here we give a derivation of equation 9 from the main text. We first calculate the first and second moments of the predictive distribution under the approximation $q(\boldsymbol{a}^L) \approx \tilde{q}(\boldsymbol{a}^L)$:

$$\mathbb{E}_{y \sim p(y)}[y] = \int yp(y|\boldsymbol{a}^L)\tilde{q}(\boldsymbol{a}^L) \, \mathrm{d}y \, \mathrm{d}\boldsymbol{a}^L$$

$$= \int\left[\int yp(y|\boldsymbol{a}^L) \, \mathrm{d}y\right]\tilde{q}(\boldsymbol{a}^L) \, \mathrm{d}\boldsymbol{a}^L$$

$$= \int m\tilde{q}(\boldsymbol{a}^L) \, \mathrm{d}\boldsymbol{a}^L$$

$$= \langle m \rangle$$

$$\mathrm{Var}[y] = \mathrm{Var}[\mathbb{E}_{y \sim p(y|\boldsymbol{a}^L)}(y)] + \mathbb{E}_{y \sim p(y)}\left[\mathrm{Var}(y|\boldsymbol{a}^L)\right]$$

$$= \mathrm{Var}[m] + \mathbb{E}_{y \sim p(y)}\left[e^\ell\right]$$

$$= \Sigma_{mm} + \int e^\ell p(y|\boldsymbol{a}^L)\tilde{q}(\boldsymbol{a}^L) \, \mathrm{d}\boldsymbol{a}^L \, \mathrm{d}y$$

$$= \Sigma_{mm} + \int e^\ell \tilde{q}(\boldsymbol{a}^L) \, \mathrm{d}\boldsymbol{a}^L$$

$$= \Sigma_{mm} + \exp\left(\langle \ell \rangle + \Sigma_{\ell\ell}/2\right)$$

where the final integral in the variance computation is performed by inserting the Gaussian form for $\tilde{q}(\boldsymbol{a}^L)$ and completing the square. Then we assume normality of the predictive distribution to obtain the result in equation 9.

### B.4 Classification: Log-likelihood

There is no exact form for the expected log-likelihood for multivariate classification with logits $\boldsymbol{a}^L$. However, using the second-order Delta method (Small, 2010), we find the expansion

$$\mathbb{E}_{\boldsymbol{a}^L \sim \tilde{q}(\boldsymbol{a}^L)}\left[\log p(y|\boldsymbol{a}^L)\right] = \left\langle \boldsymbol{a}^L \right\rangle - \mathbb{E}_{\boldsymbol{a}^L \sim \tilde{q}(\boldsymbol{a}^L)}\left[\mathrm{logsumexp}(\boldsymbol{a}^L)\right]$$

$$\approx \left\langle \boldsymbol{a}^L \right\rangle - \mathrm{logsumexp}(\langle \boldsymbol{a}^L \rangle) - \tfrac{1}{2}\left(\mathbf{p}^\top \mathrm{diag}(\boldsymbol{\Sigma}^L) - \mathbf{p}^\top\boldsymbol{\Sigma}^L\mathbf{p}\right), \quad (21)$$

To derive this expansion, we first state the second order expansion for the expectation of a function $g$ of random variable $\boldsymbol{x}$ using the Delta method as follows[8]:

$$\mathbb{E}\left[g(\boldsymbol{x})\right] \approx g\left(\mathbb{E}[\boldsymbol{x}]\right) + \tfrac{1}{2}\sum_{ij}\left[C_{ij}\frac{\partial^2 g}{\partial x_i \partial x_j}\right]_{\boldsymbol{x} = \mathbb{E}[\boldsymbol{x}]}, \qquad (22)$$

where $C_{ij} = \mathrm{Cov}(x_i, x_j)$. Now we note that the logsumexp function has a simple Hessian

$$\frac{\partial^2}{\partial x_i \partial x_j}\mathrm{logsumexp}(\boldsymbol{x}) = \delta_{ij}p_i - p_ip_j,$$

where $\boldsymbol{p} = \mathrm{softmax}(\boldsymbol{x})$. Putting these results together allows us to write:

$$\mathbb{E}\left[\mathrm{logsumexp}(\boldsymbol{x})\right] \approx \mathrm{logsumexp}\left(\mathbb{E}[\boldsymbol{x}]\right) + \tfrac{1}{2}\left[\boldsymbol{p}^\top \mathrm{diag}(\boldsymbol{C}) - \boldsymbol{p}^\top\boldsymbol{C}\boldsymbol{p}\right]_{\boldsymbol{x} = \mathbb{E}[\boldsymbol{x}]},$$

This result is sufficient to complete the derivation of equation 21 and enable training of a classifier using our method.

---

[8]This result is obtained by Taylor expansion inside the expectation.

## B.5 CLASSIFICATION: POSTERIOR PREDICTIVE DISTRIBUTION

Using the same second-order Delta method, we find the following expansion for the posterior predictive distribution:

$$p(y) = \mathbb{E}_{\boldsymbol{a}^L \sim \tilde{q}(\boldsymbol{a}^L)} \left[ p(y|\boldsymbol{a}^L) \right] \approx \mathbf{p} \odot \left[ 1 + \mathbf{p}^\top \boldsymbol{\Sigma}^L \mathbf{p} - \boldsymbol{\Sigma}^L \mathbf{p} + \tfrac{1}{2} \operatorname{diag}(\boldsymbol{\Sigma}^L) - \tfrac{1}{2} \mathbf{p}^\top \operatorname{diag}(\boldsymbol{\Sigma}^L) \right]. \tag{23}$$

where $\mathbf{p} = \operatorname{softmax}(\langle \boldsymbol{a}^L \rangle)$.

For this expansion, we begin by computing the Hessian:

$$\nabla_i \nabla_j [\boldsymbol{p}]_k = \tfrac{\partial^2}{\partial x_i \partial x_j} p_k = \left[ 2 p_i p_j - (\delta_{ki} p_j + \delta_{kj} p_i) + \delta_{ik} \delta_{jk} - \delta_{ij} p_i \right] p_k,$$

where $\boldsymbol{p} = \operatorname{softmax}(\boldsymbol{x})$, and we used the intermediate result $\nabla_j [\boldsymbol{p}]_k = \delta_{jk} p_k - p_j p_k$. Then we can form the product:

$$\operatorname{Tr} \left[ \boldsymbol{C} \nabla \nabla \boldsymbol{p} \right] = \boldsymbol{p} \odot \left[ 2 \mathbf{p}^\top \boldsymbol{C} \mathbf{p} - 2 \boldsymbol{C} \mathbf{p} + \operatorname{diag}(\boldsymbol{C}) - \mathbf{p}^\top \operatorname{diag}(\boldsymbol{C}) \right]$$

and insert this into equation 22 to obtain equation 23.

Preliminary experiments show that good results are obtained either using these approximations or a lightweight MC approximation just to perform the mapping of $\boldsymbol{a}^L$ to $(\log)\mathbf{p}$ after the deterministic heavy-lifting of computing $\boldsymbol{a}^L$. In this work we are primarily concerned with demonstrating the benefits of the moment propagation method from section 3.1, so we limit our experiments to regression examples without additional complication from approximation of the likelihood function.

## C  DEEPER NETWORKS

Here we consider the applicability of our method to the regime of deep, narrow networks. This regime is challenging because for small hidden dimension the Gaussian approximation for $\boldsymbol{a}$ (reliant on the CLT) breaks down, and these errors accumulate as the net becomes deep. We empirically explore this potential problem by investigating deep networks containing 5 layers of only 5, 25 or 125 units each. Figure 6 shows results analogous to figure 3 that qualitatively illustrate how well our approximation matches the true variational distribution of output activations both at the start and end of training. We see that our CLT-based approximation is good in the 125- and 25-unit cases, but is poor in the 5-unit case. Since it is generally considered that optimization of neural networks only works well in the high dimensional setting with at least a few tens of hidden units, these empirical observations suggest that our approximation is applicable in practically relevant architectures.

### C.1  SKIP CONNECTIONS

Training deep networks is considered difficult even in the traditional maximum-likelihood setting due to the problems of exploding and vanishing gradients. A popular approach to combat these issues is to add skip connections to the architecture. Here we derive the necessary results to add skip connections to our deterministic BNN.

We consider a simple layer with skip connections of the following form:

$$\boldsymbol{h} = f(\boldsymbol{a}'),$$
$$\boldsymbol{\delta} = \boldsymbol{h} \boldsymbol{W} + \boldsymbol{b},$$
$$\boldsymbol{a} = \boldsymbol{a}' + \boldsymbol{\delta}.$$

The moment propagation expressions for this layer are (using the bilinearity of Cov):

$$\langle a_i \rangle = \langle a_i' \rangle + \langle \delta_i \rangle,$$
$$\operatorname{Cov}(a_i, a_k) = \operatorname{Cov}(a_i', a_k') + \operatorname{Cov}(\delta_i, \delta_k) + \operatorname{Cov}(a_i', \delta_k) + \operatorname{Cov}(\delta_i, a_k'),$$

where $\langle \delta_i \rangle$ and $\operatorname{Cov}(\delta_i, \delta_k)$ can be computed using analogy to equations 14 and 15. This just leaves computation of $\operatorname{Cov}(a_i', \delta_k)$ and its transpose, which can be performed analytically using integral

results and methods borrowed from appendix A.

$$\mathrm{Cov}(a_i', \delta_k) = \mathrm{Cov}(a_i', \sum_j f(a_j')W_{jk}) + \mathrm{Cov}(a_i', b_k)$$

$$= \sum_j \left( \langle a_i' f(a_j') \rangle - \langle a_i' \rangle \langle f(a_j') \rangle \right) \langle W_{jk} \rangle$$

$$= \sum_j \langle W_{jk} \rangle \left( \Sigma_{ij}' \right) \begin{cases} (\Sigma_{jj}')^{-1}\phi(\mu_j') & \text{Heaviside} \\ \Phi(\mu_j') & \text{ReLU} \end{cases}$$

Using this result, we implement a 5-layer, 25-unit network with skip connections. In figure 6(d) we qualitatively verify the validity of our approximation on this architecture by observing a good match with Monte Carlo simulations using 20k samples.

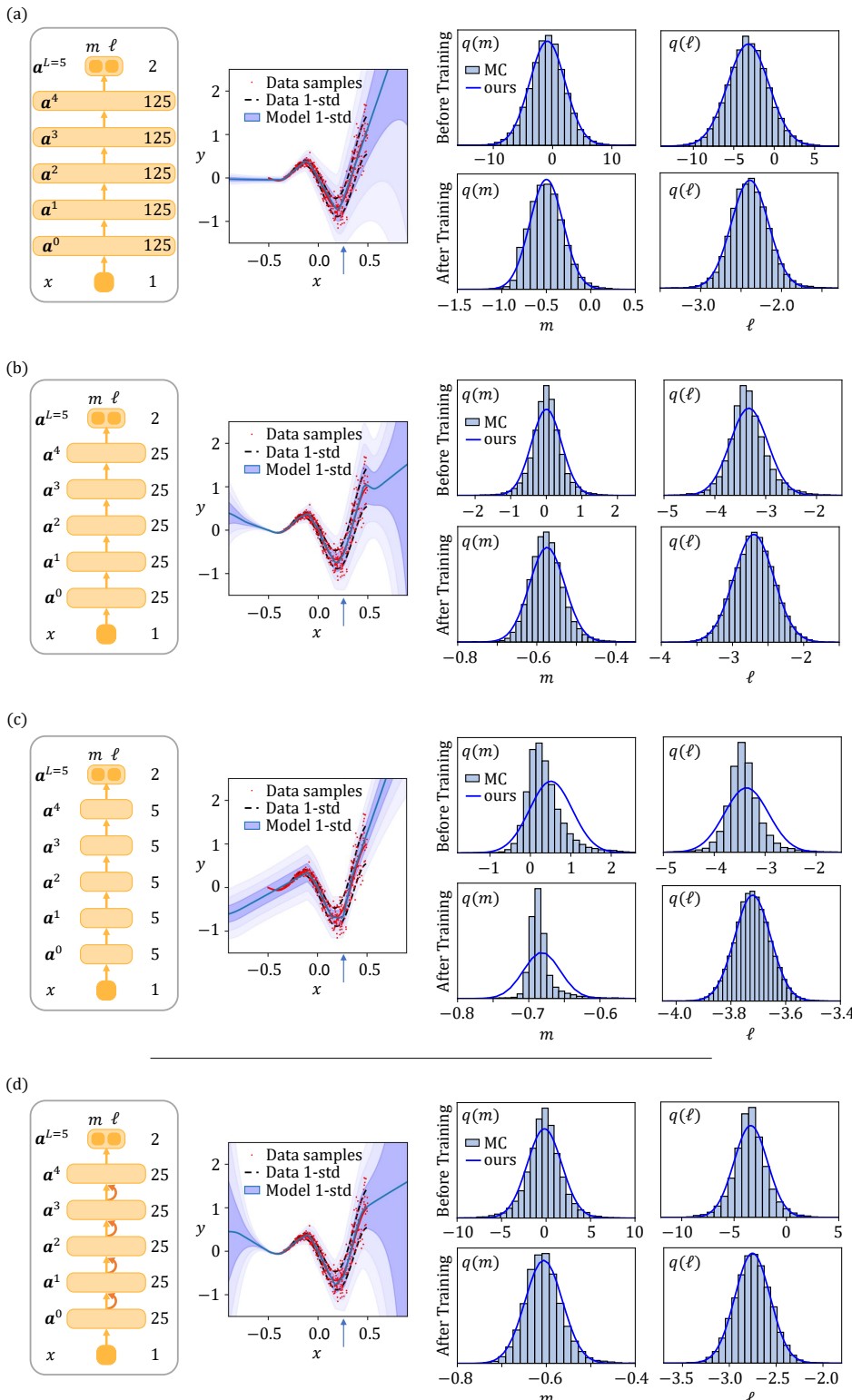

Figure 6: Empirical accuracy of our approximation for 5-layer networks trained analogously to figure 3. Progressively narrower networks of (a) 125 unit (b) 25 unit and (c) 5 unit are trained and our CLT-based approximation is only seen to significantly break down in the 5-unit case. (d) Qualitative verification of our approximation applied to an architecture with skip connections (orange).

# D    EXTENDED RESULTS

Here we include comparison with a number of different models and inference schemes on the 9 UCI datasets considered in the main text. We report test log-likelihoods at convergence and find that our method is competitive or superior to a range of state-of-the-art techniques (reproduced from Bui et al. (2016)).

| Dataset | bost | conc | ener | kin8 | nava | powe | prot | wine | yach |
|---|---|---|---|---|---|---|---|---|---|
| $|\mathcal{D}|$ | 506 | 1030 | 768 | 8192 | 11934 | 9568 | 45730 | 1588 | 308 |
| $d_x$ | 13 | 8 | 8 | 8 | 16 | 4 | 9 | 11 | 6 |
| GP 50 | $-2.22 \pm 0.07$ | $-2.85 \pm 0.02$ | $-1.29 \pm 0.01$ | $1.31 \pm 0.01$ | $4.86 \pm 0.04$ | $-2.66 \pm 0.01$ | $-2.95 \pm 0.05$ | $-0.67 \pm 0.01$ | $-1.15 \pm 0.03$ |
| DGP-1 50 | $-2.33 \pm 0.06$ | $-3.13 \pm 0.03$ | $-1.32 \pm 0.03$ | $0.68 \pm 0.07$ | $3.60 \pm 0.33$ | $-2.81 \pm 0.01$ | $-2.55 \pm 0.03$ | $-0.35 \pm 0.04$ | $-1.39 \pm 0.14$ |
| DGP-2 50 | $-2.17 \pm 0.10$ | $-2.61 \pm 0.02$ | $-0.95 \pm 0.01$ | $1.79 \pm 0.02$ | $4.77 \pm 0.32$ | $-2.58 \pm 0.01$ | $-2.11 \pm 0.04$ | $-0.10 \pm 0.03$ | $-0.99 \pm 0.07$ |
| DGP-3 50 | $-2.09 \pm 0.07$ | $-2.63 \pm 0.03$ | $-0.95 \pm 0.01$ | $1.93 \pm 0.01$ | $5.11 \pm 0.23$ | $-2.58 \pm 0.01$ | $-2.03 \pm 0.07$ | $-0.13 \pm 0.02$ | $-0.94 \pm 0.05$ |
| GP 100 | $-2.16 \pm 0.07$ | $-2.65 \pm 0.02$ | $-1.11 \pm 0.02$ | $1.68 \pm 0.01$ | $5.51 \pm 0.03$ | $-2.55 \pm 0.01$ | $-2.52 \pm 0.07$ | $-0.57 \pm 0.01$ | $-1.26 \pm 0.03$ |
| DGP-1 100 | $-2.37 \pm 0.10$ | $-2.92 \pm 0.03$ | $-1.21 \pm 0.02$ | $1.09 \pm 0.04$ | $3.75 \pm 0.37$ | $-2.67 \pm 0.02$ | $-2.18 \pm 0.06$ | $0.07 \pm 0.03$ | $-1.34 \pm 0.10$ |
| DGP-2 100 | $\mathbf{-2.09 \pm 0.06}$ | $\mathbf{-2.43 \pm 0.02}$ | $\mathbf{-0.90 \pm 0.01}$ | $2.31 \pm 0.01$ | $5.13 \pm 0.27$ | $-2.39 \pm 0.02$ | $-1.51 \pm 0.09$ | $\mathbf{0.37 \pm 0.02}$ | $-0.96 \pm 0.06$ |
| DGP-3 100 | $-2.13 \pm 0.09$ | $-2.44 \pm 0.02$ | $-0.91 \pm 0.01$ | $2.46 \pm 0.01$ | $5.78 \pm 0.05$ | $-2.37 \pm 0.02$ | $-1.32 \pm 0.06$ | $0.25 \pm 0.03$ | $-0.80 \pm 0.04$ |
| VI(KW)-2 | $-2.64 \pm 0.02$ | $-3.07 \pm 0.02$ | $-1.89 \pm 0.07$ | $\mathbf{2.91 \pm 0.10}$ | $6.10 \pm 0.19$ | $\mathbf{-2.28 \pm 0.02}$ | $\mathbf{-0.42 \pm 0.31}$ | $-0.85 \pm 0.01$ | $-1.92 \pm 0.03$ |
| SGLD-2 | $-2.38 \pm 0.06$ | $-3.01 \pm 0.03$ | $-2.21 \pm 0.01$ | $1.68 \pm 0.00$ | $3.21 \pm 0.02$ | $-2.61 \pm 0.01$ | $-1.23 \pm 0.01$ | $0.14 \pm 0.02$ | $-3.23 \pm 0.03$ |
| SGLD-1 | $-2.40 \pm 0.05$ | $-3.08 \pm 0.03$ | $-2.39 \pm 0.01$ | $1.28 \pm 0.00$ | $3.33 \pm 0.01$ | $-2.67 \pm 0.00$ | $-3.11 \pm 0.02$ | $-0.41 \pm 0.01$ | $-2.90 \pm 0.02$ |
| HMC-1 | $-2.27 \pm 0.03$ | $-2.72 \pm 0.02$ | $-0.93 \pm 0.01$ | $1.35 \pm 0.00$ | $\mathbf{7.31 \pm 0.00}$ | $-2.70 \pm 0.00$ | $-2.77 \pm 0.00$ | $-0.91 \pm 0.02$ | $-1.62 \pm 0.01$ |
| PBP-1 | $-2.57 \pm 0.09$ | $-3.16 \pm 0.02$ | $-2.04 \pm 0.02$ | $0.90 \pm 0.01$ | $3.73 \pm 0.01$ | $-2.84 \pm 0.01$ | $-2.97 \pm 0.00$ | $-0.97 \pm 0.01$ | $-1.63 \pm 0.02$ |
| VI(G)-1 | $-2.90 \pm 0.07$ | $-3.39 \pm 0.02$ | $-2.39 \pm 0.03$ | $0.90 \pm 0.01$ | $3.73 \pm 0.12$ | $-2.89 \pm 0.01$ | $-2.99 \pm 0.01$ | $-0.98 \pm 0.01$ | $-3.44 \pm 0.16$ |
| VI(KW)-1 | $-2.43 \pm 0.03$ | $-3.04 \pm 0.02$ | $-2.38 \pm 0.02$ | $2.40 \pm 0.05$ | $5.87 \pm 0.29$ | $-2.66 \pm 0.01$ | $-1.84 \pm 0.07$ | $-0.78 \pm 0.02$ | $-1.68 \pm 0.04$ |
| Dropout-1 | $-2.46 \pm 0.25$ | $-3.04 \pm 0.09$ | $-1.99 \pm 0.09$ | $0.95 \pm 0.03$ | $3.80 \pm 0.05$ | $-2.89 \pm 0.01$ | $-2.80 \pm 0.05$ | $-0.93 \pm 0.06$ | $-1.55 \pm 0.12$ |
| DVI | $-2.41 \pm 0.02$ | $-3.06 \pm 0.01$ | $-1.01 \pm 0.06$ | $1.13 \pm 0.00$ | $6.29 \pm 0.04$ | $-2.80 \pm 0.00$ | $-2.85 \pm 0.01$ | $-0.90 \pm 0.01$ | $\mathbf{-0.47 \pm 0.03}$ |
| dDVI | $-2.42 \pm 0.02$ | $-3.07 \pm 0.02$ | $-1.06 \pm 0.06$ | $1.13 \pm 0.00$ | $6.22 \pm 0.06$ | $-2.80 \pm 0.00$ | $-2.84 \pm 0.01$ | $-0.91 \pm 0.02$ | $-0.47 \pm 0.03$ |
| DVI-MC | $-2.41 \pm 0.02$ | $-3.05 \pm 0.01$ | $-1.00 \pm 0.06$ | $1.13 \pm 0.00$ | $6.25 \pm 0.03$ | $-2.80 \pm 0.00$ | $-2.85 \pm 0.01$ | $-0.93 \pm 0.04$ | $-0.55 \pm 0.03$ |
| DVI-MC-softplus | $-2.42 \pm 0.02$ | $-3.06 \pm 0.02$ | $-1.03 \pm 0.05$ | $1.13 \pm 0.00$ | $6.20 \pm 0.04$ | $-2.80 \pm 0.01$ | $-2.85 \pm 0.01$ | $-0.89 \pm 0.01$ | $-0.54 \pm 0.03$ |
| MCVI | $-2.46 \pm 0.02$ | $-3.07 \pm 0.01$ | $-1.03 \pm 0.04$ | $1.14 \pm 0.00$ | $5.94 \pm 0.05$ | $-2.80 \pm 0.00$ | $-2.87 \pm 0.01$ | $-0.92 \pm 0.01$ | $-0.68 \pm 0.03$ |
| MCVI-softplus | $-2.47 \pm 0.02$ | $-3.08 \pm 0.02$ | $-1.02 \pm 0.05$ | $1.14 \pm 0.00$ | $5.99 \pm 0.02$ | $-2.80 \pm 0.00$ | $-2.85 \pm 0.01$ | $-0.92 \pm 0.01$ | $-0.69 \pm 0.03$ |

Table 3: Average test log likelihood on UCI datasets. See table 4 for model glossary and implementation references.

| Abbreviation | Description | Implementation |
|---|---|---|
| GP N | Gaussian process regression | Bui et al. (2016) |
| DGP-L N | Deep Gaussian process regression | Bui et al. (2016) |
| VI(KW)-L | BNN with the variational free energy evaluated using the reparameterization trick (KW = Kingma + Welling) | Bui et al. (2016) |
| SGLD-L | Stochastic Gradient Langevin Dynamics | Bui et al. (2016) |
| HMC-L | Hamiltonian Monte Carlo | Bui et al. (2016) using toolbox Vanhatalo & Vehtari (2006) |
| PBP-L | probabilistic back-propagation | Hernández-Lobato & Adams (2015) |
| VI(G)-L | scalable variational inference (VI) method for neural networks (G = Graves) | Graves (2011) |
| Dropout-L | a technique that employs dropout during training as well as at prediction time. | Gal & Ghahramani (2016) |
| DVI | Our method | **ours** |
| dDVI | same as DVI, but with diagonal activation covariance | **ours** |
| DVI-MC | same as DVI, but with a light-weight Monte Carlo integration only for computing the predictive distribution | **ours** |
| DVI-MC-softplus | same as DVI-MC, but uses $\mathrm{softplus}(\ell)$ to model heteroscedastic observation variance rather than $e^\ell$. Note that in this case there is no closed form for the log-likelihood, so the lightweight final MC step is required | **ours** |
| MCVI-exp | our implementation of SVI using $e^\ell$ | **ours** |
| MCVI-softplus | our implementation of SVI using $\mathrm{softplus}(\ell)$ | **ours** |

Table 4: Glossary of methods displayed in table 3 with references. Note: L = number of hidden layers; N = number of Gaussian process pseudo-points. Please refer to Bui et al. (2016) for more descriptions of other state-of-the-art methods.

## D.1 ABLATION STUDY

Here we provide an ablation study that indicates the individual contributions of (1) the deterministic approximation and (2) the the empirical Bayes prior. We consider all combinations of DVI or MCVI with and without empirical Bayes. In the DVI-fixed and MCVI-fixed cases without empirical Bayes we use a fixed zero-mean Gaussian prior during training and we perform separate runs to tune the prior variance, reporting the best performance achieved (cf. figure 5)[9]. Since the EB approach requires no hyperparameter tuning between the datasets shown, these results hide the considerable computational advantaged that the EB approach brings.

## E   VARIANCE REDUCTION AND THE LOCAL REPARAMETERIZATION TRICK

By eliminating MC sampling and its associated variance entirely, our method directly tackles the problem of high variance gradient estimates that hinder MC approaches to training of BNNs. Alternative methods that only *reduce* variance have been considered, and among these, the local

---

[9]Note that the EB method can out perform manual tuning because it automatically finds different prior variances for each weight matrix, whereas manual tuning case searches over a single hyperparameter controlling all prior variances.

| Dataset | DVI | DVI-fixed | MCVI | MCVI-fixed |
|---|---|---|---|---|
| bost | $-2.41 \pm 0.02$ | $-2.46 \pm 0.02$ | $-2.46 \pm 0.02$ | $-2.48 \pm 0.02$ |
| conc | $-3.06 \pm 0.01$ | $-3.07 \pm 0.01$ | $-3.07 \pm 0.01$ | $-3.07 \pm 0.01$ |
| ener | $-1.01 \pm 0.06$ | $-1.07 \pm 0.04$ | $-1.03 \pm 0.04$ | $-1.07 \pm 0.04$ |
| kin8 | $1.13 \pm 0.00$ | $1.12 \pm 0.00$ | $1.14 \pm 0.00$ | $1.13 \pm 0.00$ |
| nava | $6.29 \pm 0.04$ | $6.32 \pm 0.04$ | $5.94 \pm 0.05$ | $6.00 \pm 0.02$ |
| powe | $-2.80 \pm 0.00$ | $-2.80 \pm 0.01$ | $-2.80 \pm 0.00$ | $-2.80 \pm 0.00$ |
| prot | $-2.85 \pm 0.01$ | $-2.84 \pm 0.01$ | $-2.87 \pm 0.01$ | $-2.89 \pm 0.01$ |
| wine | $-0.90 \pm 0.01$ | $-0.94 \pm 0.01$ | $-0.92 \pm 0.01$ | $-0.94 \pm 0.01$ |
| yach | $-0.47 \pm 0.03$ | $-0.49 \pm 0.03$ | $-0.68 \pm 0.03$ | $-0.56 \pm 0.03$ |

Table 5: Ablation study of all combinations of DVI and MCVI with EB or a fixed prior. One standard deviation error in the last significant digit is shown in paarentheses.

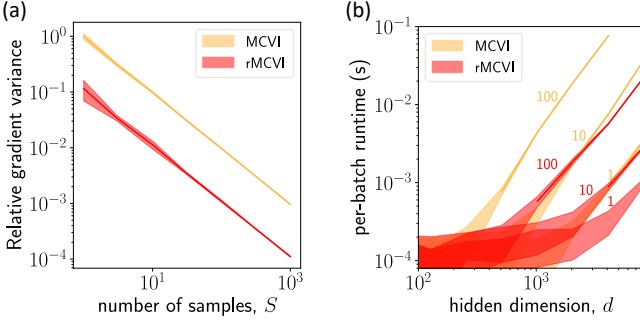

Figure 7: Performance of MCVI vs rMCVI. (a) Gradient variance for the model shown in figure 3 with batch size $B = 1$. Variance values are normalized such that MCVI with 1 sample appears at unit relative variance. For this model, rMCVI achieves the same variance as MCVI with roughly $5\times$ fewer samples (b) Runtime performance of rMCVI evaluated under the conditions of figure 4. For this model, rMCVI runs with roughly $10\times$ more samples in the same time as MCVI.

reparameterization trick (Kingma et al., 2015) is particularly popular. Similar to our approach, the local reparameterization trick maps the uncertainty in the weights to an uncertainty in activations, however, unlike the fully deterministic DVI, MC methods are then used to propagate this uncertainty through non-linearities. The benefits of MCVI with the reparameterization trick (rMCVI) over vanilla MCVI are two-fold:

- The variance of the gradient estimates during back propagation are reduced (see details in Kingma et al. (2015)).
- Since the sampling dimension in rMCVI only appears on the activations and not on the weights, an $H' \times H$ linear transform can be implemented using $SB \times H'$ by $H' \times H$ matrix multiplies (where $S$ is the number of samples and $B$ is the batch size). This contrasts with the $S \times B \times H'$ by $S \times H' \times H$ batched matrix multiply required for MCVI. Although both of these algorithms have the same asymptotic complexity $\mathcal{O}(SBH'H)$, a single large matrix multiplication is generally more efficient on GPUs than smaller batched matrix multiplies.

Figure 7 shows empirical studies of the gradient variance and runtime for rMCVI vs. MCVI applied to the model described in section 3.1.1 and figure 3. To evaluate the gradient variance, we initialize the model with partially trained weights and measure the variance of the gradient of the ELBO reconstruction term $\mathcal{L}$ with respect to variational parameters. Specifically, we inspect the gradient with respect to the parameters $\boldsymbol{\Sigma}_q^L$ describing the variance of the $q$ distribution for the weight matrix in the final layer.

$$\text{Gradient variance} := \underset{s \in \boldsymbol{\Sigma}_q^L}{\text{mean}} \left[ \text{Var} \left( \frac{\partial \mathcal{L}}{\partial s} \right) \right]$$

The plots in figure 7 serve to show that rMCVI is not fundamentally different from MCVI, and the performance of one (on either the speed or variance metric) can be transformed into the other by varying the number of samples. A comparison of DVI with rMCVI is included in table 3 using the implementation labelled as "VI(KW)-1".

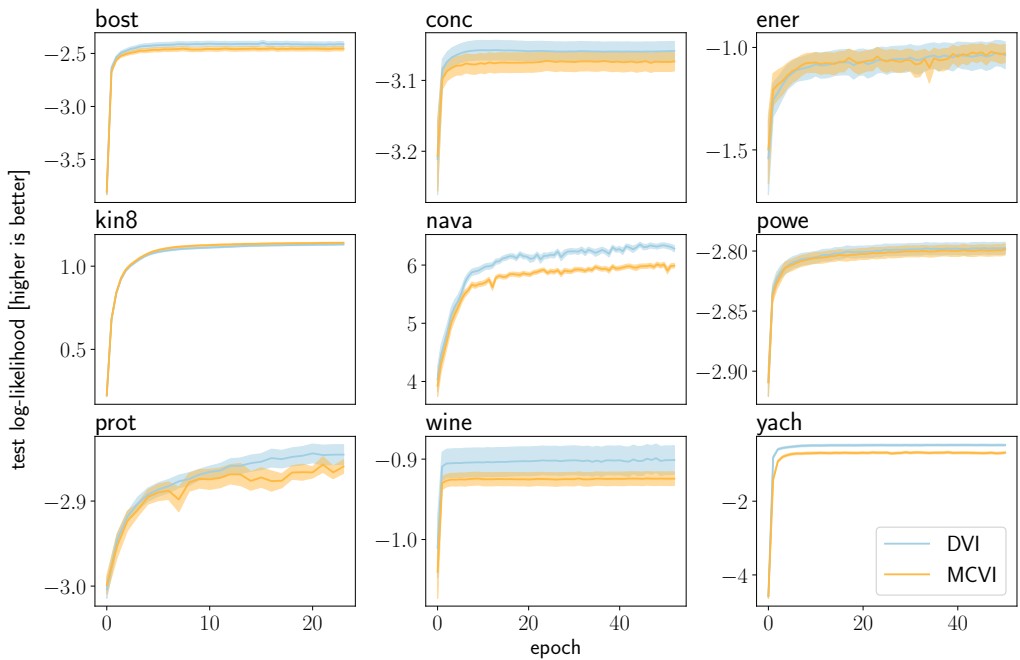

Figure 8: Learning trajectories for the models from table 2.

## F    LEARNING CURVES

Figure 8 shows the test log-likelihood during the training of the models from table 2 using DVI and MCVI inference algorithms. Since the underlying model is identical, both methods should achieve the same test log-likelihood given infinite time and infinite MC samples (or a suitable learning rate schedule) to mitigate the increased variance of the MCVI method. However, since we use only 10 samples and do not employ a leaning rate schedule, we find that MCVI converges to a log-likelihood that is consistently worse than that achieved by DVI.

