# OpenReview forum: "Deterministic Variational Inference for Robust Bayesian Neural Networks"
_ICLR.cc/2019/Conference_

### Official Review · AnonReviewer2 · 2018-10-25
**Two advances for variational Bayes on neural networks. Expectations are done deterministically (as in PBP), not by Monte Carlo, thus reducing variance. The weight prior is learned with length scales by empirical Bayes. Both should make VB training more robust, but experiments do not show that.**

**Rating:** 7
**Confidence:** 5

**Review:**

Summary:

This work is tackling two difficulties in current VB applied to DNNs ("Bayes by backprop"). First, MC approximations of intractable expectations are replaced by deterministic approximations. While this has been done before, the solution here is new and very interesting. Second, a Gaussian prior with length scales is learned by VB empirical Bayes alongside the normal training, which is also very useful.

The term "fixing VB" and some of the intro is not really supported by the rather weak experiments, done on small datasets and networks, where much older work like Barber&Bishop would apply without any problems. While interesting and potentially very useful novelties are presented, and the writing is excellent, both experiments and motivation can be improved.

- Quality: Extremely well written paper, I learned a lot from it. Approximations are
   tested, great figures to explain things. And the major technical novelty, the
   expression for <h_j h_l>, is really interesting and useful.
- Clarity: Excellent writing until it comes to the experiments. Here, important
   details are just missing, for example what q(w) is (fully factorized Gaussian?).
   Very nice literature review, also historical.
- Originality: The idea of matching Gaussian moments along the network graph is
   previously done in PBP (Lobato, Adams), as acknowledged here. Porting this from
   ADF to VB gives dDVI. PBP also has the property that a DL system gives you the
   gradients. Having said that, I think dDVI may be more useful than PBP.
   While Barber&BIshop 98 is cited, they miss the expression for <h_j h_l> in
   there. Now, what is done here, is more elegant, does not need 1D quadrature.
- Significance: Judging from the existing experiments, the significance may be
   rather small, *if one only looks at test log likelihood*. I'd still give this the
   benefit of the doubt, as in particular dDVI could be really interesting at large
   scale as well. But the authors may tone down their language a bit.
   To increase significance, I recommend to comment beyond just test log
   likelihood scores. For example:
   - Does the optimization become simpler, less tuning required, more automatic?
      Would one not expect so, given you make a big point out of reducing variance?
      Does it converge faster?
   - Can you do something with your posterior that normal DNN methods cannot
      do? Better decisions (bandits, active learning, HPO)? Continual learning?
      In the end, who really cares about test log likelihood?

Experiments:
- What is the q(w) family being used here? Fully factorized Gaussian? I
   suppose so for dDVI. But for DVI? Not said anywhere, in main paper or
   Appendix
- A bit disappointing. Why not evaluate at least dDVI with diagonal q(w) on
   some much larger models and datasets? Why not quote numbers on speed
   and robustness of learning, etc? Show what you really gain by reducing the
   variance.
- Experiments are OK, but on pretty small datasets, and for single hidden
   layer NNs. On such data and models, the Barber&Bishop 98 method could
   be run as well
- Was MCVI run with re-parameterization? This is really important. If not,
   this would be an important missing comparison. Please be clear in the main
   text
- Advantages over MCVI are not very large. At least, dDVI should be faster to
   converge than MCVI.
   Can you say something about robustness of training? Is it easier to train
   dDVI than MCVI?
- Why not show the PBP-1 results, comparing to dDVI, in the main text? Are they
   obtained with the same model? dDVI is doing better.

Other points:
- Please acknowledge the <h_j h_l> expression in Barber&Bishop 98. Yours is
   more elegant and faster (does not need 1D quadrature)
- Relation to PBP: Note that dDVI has an advantage in practice. With PBP, I need
   to compute gradients for every datapoint. In dDVI, I can do mini-batch
   updates.
- I just *love* the header "Wild approximations". I tend to refer to this kind of work
   as "weak analogies". Why do you not also compare against this, and show it really
   does not work?

---

> ### Author Response · Authors · 2018-11-19
> **Response to reviewer questions**
>
> Thank you for your detailed and enthusiastic review. We have updated the paper and address specific questions below
>
> > The term "fixing VB" and some of the intro is not really supported… the authors may tone down their language a bit.
>
> We have removed “fixing VB” from the title and removed strong phrases in the abstract and introduction.
>
> > While Barber&BIshop 98 is cited, they miss the expression for <h_j h_l> in there. Now, what is done here, is more elegant, does not need 1D quadrature.
>
> We have added these comments to our related work section.
>
> > Can you do something with your posterior that normal DNN methods cannot do?
>
> Standard training of DNNs which returns point-parameter estimates (i) results in poorly calibrated predictive uncertainty estimates (notably predictions are often confidently wrong), (ii) does not support model-based sequential decision making (e.g. active learning), and (iii) suffers from catastrophic forgetting when trained in the continual learning setting. BNNs have been shown to substantially improve upon standard models / training in these three settings (see e.g. [1,2,3]). The new innovations proposed in this paper will be applied to these areas in future work. In the second two application areas – sequential decision making and continual learning – approximate Bayesian inference must be run as an inner loop of a larger algorithm. This requires a robust and automated version of BNN training: this is precisely where we believe the innovations in this paper will have large impact since they pave the way to automated and robust deployment of BBNs that do not involve an expert in-the-loop. We have included these points as motivational future work items in the paper conclusion.
>
> > what is q(w) [the variational family used in the experiments]?
>
> Our method is not limited to fully factorized Gaussian variational distributions (any distribution with a tractable first and second moment could be used). However, for computational simplicity, our experiments do use a fully factorized Gaussian q(w). We have added this detail to the experimental section.
>
> > Why not evaluate at least dDVI with diagonal q(w) on some much larger models and datasets?
>
> We have added an appendix C that evaluates the performance of DVI in larger models including deep networks with skip connections. Regarding larger datasets, our evaluation focuses on assessing the robustness of the new methods and how automatic they are. The experiments do consider nine different datasets, following established practice for evaluating new approximate inference methods for BNNs (see e.g. [4,5,6]). We evaluate the proposed methods using many of different model variants (hetero vs homoscedastic, MC vs different deterministic approximations, different prior settings, various methods for parameterising the variance, etc.). In this way we have prioritized a comprehensive assessment of the myriad design decisions, rather than assessing a relatively small number of design decisions on a larger number of datasets. Whilst we acknowledge that since the benchmarks are relatively simple this work is just a first step of a completely comprehensive evaluation, we believe that the experiments provide a solid foundation for this longer-term enterprise.
>
> > Was MCVI run with re-parameterization?
>
> We run vanilla MCVI, and re-parameterization is discussed in section E of the appendix and results using re-parameterization appear in Table 3. We have added this clarification and pointers to section E in the main text.
>
> > Relation to PBP: Note that dDVI has an advantage in practice… Why not show the PBP-1 results, comparing to dDVI, in the main text? Are they obtained with the same model? dDVI is doing better.
>
> Table 3 is too large to be included in the main text and although we perform comparison with PBP using the same model, we don’t want to move the results to the main text because our method has clear qualitative advantages over PBP as you highlight: 1) we handle batches of data and do not have to process one data point at a time, 2) we account for correlations in the forward pass and in the posterior distribution and 3) we can account for heteroskedastic noise. We have clarified these advantages in the related work section.
>
> > Compare against [dropout-like methods], and show it really does not work?
>
> Our extended results table (Table 3) in the appendix includes results using dropout.
>
> [1] Known Unknowns: Uncertainty Quality in BNNs, R Oliveira et al., NIPS BDL workshop 2016
> [2] Deep Bayesian Active Learning with Image Data, Y Gal et al., PMLR 70:1183-1192, 2017
> [3] Variational Continual Learning CV Nguyen et al. ICLR 2018
> [4] Deep Gaussian Processes for Regression using Approximate Expectation Propagation. T Bui, et al. ICML 2016
> [5] Black-box alpha-divergence minimization JM Hernández-Lobato et al., ICML 2016
> [6] Simple and Scalable Predictive Uncertainty Estimation using Deep Ensembles B Lakshminarayanan, et al., NIPS 2017

---

### Official Review · AnonReviewer1 · 2018-10-30
**Fixing Variational Bayes: Deterministic Variational Inference for Bayesian Neural Networks**

**Rating:** 7
**Confidence:** 3

**Review:**

This paper considers a purely deterministic approach to learning variational posterior approximations for Bayesian neural networks.  Variational lower bound gradients are obtained by approximating the lower bound using Gaussian approximations and moment propagation for network activations, and using a closed form expression for the variational expectation of the log-likelihood, the latter being available for the models considered in the paper.

This is an interesting paper.  The Gaussian approximations and moment propagation approximations are clever and highly original although the derivation is rather heuristic.  There is some empirical support that the approximations work well.  The paper is generally well written and clearly motivated in the context of the existing literature.

The approximations work well for the examples presented in the paper.  The experiments are for rather small datasets and for the DVI method if I understand correctly only models with a single hidden layer are considered.  I wonder if the Gaussian and moment propagation approximations cause difficulty when applied repeatedly in deeper networks.  Are the problems with MCVI and high gradient variance most serious for large datasets and more complex models?  If so a comparison of DVI with MCVI in a more complex example is of interest.  The empirical Bayes approximations are interesting - I would have thought similar approximations been used in the literature before, in addition to the work you mention in Section 5?  I don't feel there is much to compare the proposed EB approximations to, although a comparison with manual tuning is given in Section 6.

---

> ### Author Response · Authors · 2018-11-19
> **Response to review questions**
>
> Thank you for your kind review and suggestions for additional studies. We address specific questions below with reference to new sections in the paper.
>
> > I wonder if the Gaussian and moment propagation approximations cause difficulty when applied repeatedly in deeper networks
>
> We have added a new Appendix C that studies deeper neural networks. We also include a 5-layer, 125-unit network (“deep, wide”), 5-layer, 25-unit network (“deep, narrow”) and a 5-layer, 5-unit network (“deep and impractically narrow”). For the practically relevant 25- and 125-unit cases, we observe good fits to data and qualitatively good agreement of our approximation with Monte Carlo (MC) simulation using 20k samples. For the extremely narrow 5-unit case we see significantly non-Gaussian output distributions in MC simulation due to failure of the central limit theorem underlying our approximations. These experiments cover the range of behaviour expected in our method and demonstrate that our method does work for deep networks in the practically relevant regime where at least a few tens of hidden units are used. In addition, we have derived the required results to incorporate skip connections into our method to help training on even deeper networks. All of these results are summarized in figure 6.
>
> > Are the problems with MCVI and high gradient variance most serious for large datasets and more complex models? If so a comparison of DVI with MCVI in a more complex example is of interest.
>
> Our evaluation focuses on assessing the robustness of the new methods and how automatic they are. The experiments do consider nine different datasets (containing up to 45k examples), in accordance with established practice for evaluating new approximate inference methods for BNNs (see e.g. [1,2,3]). Crucially we evaluate the proposed methods using many different model variants (hetero vs homoscedastic, MC vs different deterministic approximations, different prior settings, various methods for parameterising the variance, etc.). In this way we have prioritized a comprehensive assessment of the myriad design decisions, rather than assessing a relatively small number of design decisions on a larger number of datasets. Whilst we acknowledge that since the benchmarks are relatively simple this work is just a first step of a completely comprehensive evaluation, we believe that the experiments provide a solid foundation for this longer-term enterprise.
>
> > I don't feel there is much to compare the proposed EB approximations to, although a comparison with manual tuning is given in Section 6.
>
> To complement our comparison with manual tuning, we have added section D.1 and Table 5 to the appendix to give an ablation study corresponding to all combinations of DVI or MCVI with fixed or EB priors. Note that when running with a fixed prior, we select the best prior variance by a separate hyperparameter sweep on each dataset (cf. figure 5). Besides eliminating this tuning overhead, EB maintains a small performance advantage over manual tuning because it automatically finds different prior variances for each weight matrix, whereas we only manually tune the global fixed prior variance.
>
> [1] Deep Gaussian Processes for Regression using Approximate Expectation Propagation. Thang Bui, José Miguel Hernández-Lobato, Yingzhen Li, Daniel Hernández-Lobato, and Rich Turner
> ICML 2016
> [2] Black-box alpha-divergence minimization José Miguel Hernández-Lobato, Yingzhen Li, Mark Rowland, Daniel Hernández-Lobato, Thang Bui, and Rich Turner
> ICML 2016
> [3] Simple and Scalable Predictive Uncertainty Estimation using Deep Ensembles Balaji Lakshminarayanan, Alexander Pritzel, Charles Blundell, NIPS 2017

---

### Official Review · AnonReviewer3 · 2018-11-02
**An interesting paper**

**Rating:** 7
**Confidence:** 3

**Review:**

The authors propose a new approach to perform deterministic variational inference for feed-forward BNN with specific nonlinear activation functions by approximating layerwise moments. Under certain conditions, the authors show that the proposed method achieves better performance than existing Monte Carlo variational inference. This paper is interesting since most of the existing works focus on Monte Carlo variational inference. The main contribution of this paper is to perform Gaussian approximation. The authors show that for specific activation functions, the Gaussian approximation is reasonable. The main concern is the cumulative error due to the Gaussian approximation. Since the authors argue that the proposed method fixes the issues of stochastic VI for BNN, the authors should also investigate/clarify the following cases.
(1)  A deep BNN to show that the cumulative error is negligible as the number of the hidden layers increases
(2)  Small latent dimension since CLT may not hold
(3)  A heavy-tailed variational distribution since the second moment may not be finite
(4)  Other nonlinear activations since the Gaussian approximation may not be accurate due to (generalized) Berry-Esseen theorem
(5) A BNN with skip connections  since a Bayesian multiplayer perceptron with skip connections is also a feed-forward BNN

Among these cases, I am eager to see some results on a deep thin BNN. For example, a BNN with 5 hidden layers, where the latent dimension at each layer is less than 32.
Furthermore, I would like to see some empirical comparison on real-world datasets between DVI and MCVI under a *fixed* prior since such comparison demonstrates the approximation accuracy of DVI and rule out the confounding factor introduced by the empirical Bayes approach.

---

> ### Author Response · Authors · 2018-11-19
> **Revisions to address requests**
>
> Thank you for your great recommendations for additional studies. We created some new sections when replying to your questions that are good improvements to the paper:
>
> > (1) A deep BNN to show that the cumulative error is negligible as the number of the hidden layers increases
> > (2) Small latent dimension since CLT may not hold … I am eager to see some results on a deep thin BNN. For example, a BNN with 5 hidden layers, where the latent dimension at each layer is less than 32
>
> We have added a new Appendix C that studies deeper neural networks:
> To address request (1), we studied a 5-layer, 125-unit network, and we observe good fits to data and qualitatively good agreement of our Gaussian approximation with Monte Carlo (MC) simulation using 20k samples (see the new figure 6a)
> To address request (2) we additionally study a 5-layer, 25-unit network, and we again see good fits and qualitatively reasonable performance of our approximation (figure 6b). For completeness, we include an extreme case: 5-layers, 5-units. In this extreme case we do see we see significantly non-Gaussian output distributions in MC simulation due to failure of the central limit theorem underlying our approximations. Since such narrow networks are not of significant practical importance, we do not see this as a major problem with our method.
> We thank the reviewer for recommending these studies – including a demonstration that failure cases only arise in impractically narrow architectures helps to justify our use of the CLT.
>
> > (3) A heavy-tailed variational distribution since the second moment may not be finite
>
> The reviewer is correct that our method relies on a variational distribution with finite first and second moments, for which the CLT holds. We have added clarification of these necessary conditions in the text. Note that *only* the first and second moments of the variational distribution are required to compute the reconstruction log-probability in the ELBO (i.e. only <W> and Cov(W,W) appear in equation 3). The precise form of the variational distribution is only required to evaluate the KL term in the ELBO, and therefore it is easy to apply our method to any variational family with finite moments which has a closed form KL with a suitable prior.
>
> > (4) Other nonlinear activations since the Gaussian approximation may not be accurate due to (generalized) Berry-Esseen theorem
>
> We provide results on the Heaviside and ReLU nonlinearities. Other useful and commonly deployed nonlinearities are (generally speaking) “softened” and translated versions of either a Heaviside or ReLU nonlinearity (e.g. tanh is a soft Heaviside and elu is a soft ReLU). Note that the nonlinearity only appears in equations 4 and 5, where it is being convolved with the Gaussian activation distribution. Since this convolution already softens the hard nonlinearities (e.g. see the smooth functions plotted in figure 2), changes in the intrinsic the softness of the underlying nonlinearity are qualitatively equivalent to using a hard nonlinearity and adjusting the convolving Gaussian covariance. For this reason, we do not think there will be considerable benefit from exploring other nonlinearities. We believe that any gain is likely not worth the considerable work required to find closed form approximations for the integrals in 4 and 5 for arbitrary nonlinearities.
>
> > (5) A BNN with skip connections since a Bayesian multiplayer perceptron with skip connections is also a feed-forward BNN
>
> This was fairly simple to add to our method, and we thank the reviewer for suggesting this nice addition. Specifically, we have added derivations of the integral results required to implement a network with skip connections in a new Appendix C.1 and include a figure showing that our approximation works in a deep network with skip connections in Fig 6d.
>
> > I would like to see some empirical comparison on real-world datasets between DVI and MCVI under a *fixed* prior since such comparison demonstrates the approximation accuracy of DVI and rule out the confounding factor introduced by the empirical Bayes approach.
>
> We have added section D.1 and Table 5 to the appendix to give an ablation study corresponding to all combinations of DVI or MCVI with fixed or EB priors. Note that when running with a fixed prior, we select the best prior variance by a separate hyperparameter sweep on each dataset (cf. figure 5). Besides eliminating this tuning overhead, EB maintains a small performance advantage over manual tuning because it automatically finds different prior variances for each weight matrix, whereas we only manually tune the global fixed prior variance.

---

### Public Comment · ~Alexander_Shekhovtsov1 · 2019-01-30
**Discussion**

Thanks for a very well done work. With Bayes by backprop we have indeed experienced that it was necessary to tune priors, to adjust the learning rate to compensate for the increased stochasticity (less robust training) and even to weight the complexity term (KL to the prior) in the case of non-iid data (dependent pixels, augmentations, etc). I would appreciate very much if the authors could comment on the following questions.

(1) The heteroskedastic model predicting mean and variance of a Gaussian distribution is a bit unclear to me. In the Bayesian predictive distribution there is variance Sigma_{mm} coming from the uncertainty of parameters and the variance e^{l} learned as a part of the model. In case of a large training sample, Sigma_{mm} approaches zero and the predictive distribution (9) to the ML plug-in. However, we are interested to see the effect of the Bayesian learning, the quality of estimating Sigma_{mm}, i.e. in the case of a small training sample. I am not sure this is the case in Fig 3. It looks like ML would give a similar predictive distribution and the Bayesian Sigma_{mm} and especially Sigma_{ll} are small corrections on top of that. Does it really show the utility of the method?
What is further confusing, is that the whole difference between Sigma_{mm} and e^{l} is that the former is not fitting the predictive model to the data whereas the later is, otherwise technically they are replaceable, at least Sigma_{mm} can implement e^{l} with some extra neurons and noisy weights.

(2) The paper mentions that "strong test performance of DVI is largely retained by dDVI" and "dDVI surprisingly retains much of the performance". However, the only experiment relevant to this is the model with one hidden layer (Table 2). The only non-zero covariance term in this case is Sigma_{m,l} for the output unit, which by the way is only appears in the expected likelihood (8) but not in the predictive probability (9). So the two methods may be expected to be nearly identical. Accounting for correlations is one main technical contribution of the work. It would be really interested to know whether such accounting for correlations is important in deeper models. And, more broadly, whether restricting the space of models to such where activations given the input are uncorrelated and learning a model in this space is in any way inferior.

(3) A follow up on the question from the referees "who cares about test likelihood". Could you suggest to researchers interested in BNNs what would be a more practically relevant and accessible to non-experts set of experiments to test the quality of the learned probabilistic model for both regression and classification? And also, related to (1), to see the value of the Bayesian part?

With kind regards,

---

### Meta-Review · Area_Chair1 · 2018-12-13

**Confidence:** 4
**Recommendation:** Accept (Oral)

**Metareview:**

The manuscript proposes deterministic approximations for Bayesian neural networks as an alternative to the standard Monte-Carlo approach. The results suggest that the deterministic approximation can be more accurate than previous methods. Some explicit contributions include efficient moment estimates and empirical Bayes procedures.

The reviewers and ACs note weakness in the breadth and complexity of models evaluated, particularly with regards to ablation studies. This issue seems to have been addressed to the reviewer's satisfaction by the rebuttal. The updated manuscript also improves references to related prior work.

Overall, reviewers and AC agree that the general problem statement is timely and interesting, and well executed. We recommend acceptance.